# Contextual fear memory retrieval by correlated ensembles of ventral CA1 neurons

Jessica C. Jimenez[1,2], Jack E. Berry [1,2,5], Sean C. Lim[1,2,5], Samantha K. Ong[1,2], Mazen A. Kheirbek [3,4] & Rene Hen [1,2✉]

Ventral hippocampal CA1 (vCA1) projections to the amygdala are necessary for contextual fear memory. Here we used in vivo $Ca^{2+}$ imaging in mice to assess the temporal dynamics by which ensembles of vCA1 neurons mediate encoding and retrieval of contextual fear memories. We found that a subset of vCA1 neurons were responsive to the aversive shock during context conditioning, their activity was necessary for memory encoding, and these shock-responsive neurons were enriched in the vCA1 projection to the amygdala. During memory retrieval, a population of vCA1 neurons became correlated with shock-encoding neurons, and the magnitude of synchronized activity within this population was proportional to memory strength. The emergence of these correlated networks was disrupted by inhibiting vCA1 shock responses during memory encoding. Thus, our findings suggest that networks of cells that become correlated with shock-responsive neurons in vCA1 are essential components of contextual fear memory ensembles.

[1] Departments of Neuroscience, Psychiatry & Pharmacology, Columbia University, New York, NY, USA. [2] Division of Integrative Neuroscience, Department of Psychiatry, New York State Psychiatric Institute, New York, NY, USA. [3] Neuroscience Graduate Program, Weill Institute for Neurosciences, Kavli Institute for Fundamental Neuroscience, Center for Integrative Neuroscience, San Franciso, CA, USA. [4] Department of Psychiatry, University of California, San Francisco, CA, USA. [5]These authors contributed equally: Jack E. Berry, Sean C. Lim. ✉email: rh95@columbia.edu

The hippocampus (HPC) is critical for the formation of contextual memories[1–6]. While the HPC has classically been considered a purely cognitive structure that processes contextual and spatial information independent of valence, a growing body of work has also supported a role for the HPC in emotional behavior[7–14]. This dual function is thought to be segregated along the dorso-ventral axis of the HPC, with the dorsal HPC involved in cognitive processes, and the ventral HPC (vHPC) involved in emotional behaviors[15,16]. The specialization of the vHPC in emotional behavior is further supported by recent work which identified vHPC representations of aversive stimuli that are necessary for anxiety-related behavior[7,9,12].

Within the vHPC, functional specialization is delineated into distinct subsets of neurons that project to specific limbic structures to mediate unique aspects of behavior[9,13,17–24]. One such functional distinction has recently been identified between the projections to the lateral hypothalamus (LHA) and basal amygdala (BA), wherein ventral hippocampal CA1 (vCA1) projections to the BA, but not LHA, are necessary for contextual fear conditioning[9]. Still, how representations of aversive stimuli may be incorporated into emotional memories within the vHPC, and the neural circuit mechanisms by which vCA1 projections to the amygdala mediate contextual fear memory formation and recall remain unknown.

## Results

### vCA1-BA projection is enriched in shock responsive neurons.
To investigate the circuit mechanisms by which vCA1 pyramidal neurons in the HPC encode and retrieve contextual fear memories, we imaged the activity of these neurons during a 3-day contextual memory paradigm. Given that activity in vCA1 projections to the BA but not LHA is necessary for contextual fear memory encoding and retrieval[9], we first assessed the activity patterns in these two cell populations to identify signatures of activity during contextual fear conditioning that are specialized to the vCA1-BA projection. We utilized a retrograde CAV2-Cre virus in the BA or LHA to selectively express Cre-dependent GCaMP6f in these projections as previously described[9], which allowed us to record and track the activity of sparse populations of vCA1-BA and vCA1-LHA projecting neurons (Fig. 1a). vCA1 activity was then recorded during exposure to a context in which mice received an aversive foot shock after a period of exploration (Fig. 1b). vCA1-BA and vCA1-LHA mice displayed freezing behavior during re-exposure to the context the following day, but not during subsequent exposure to a novel context (Fig. 1b), demonstrating selective contextual-fear memory retrieval in the conditioning context. We found that vCA1-LHA projecting neurons were significantly more active than vCA1-BA projecting neurons at baseline (Supplementary Fig. 1A, B), and exhibited a significant decrease in $Ca^{2+}$ event rate during context fear retrieval, while the $Ca^{2+}$ event rate of vCA1-BA neurons was stable across days (Fig. 1c, Supplementary Fig. 1B). Interestingly, despite the relatively low event rate of vCA1-BA compared to vCA1-LHA projecting neurons, we observed robust responses in vCA1-BA neurons during the A1 shock period (Fig. 1d), with far fewer shock-responsive neurons in the vCA1-LHA projection (Fig. 1d, e). Moreover, the population of vCA1-BA neurons (but not vCA1-LHA) that were shock-responsive during context encoding also exhibited shock responses when mice were treated with additional shocks during subsequent tone–shock pairings (Supplementary Fig. 1C, D). Finally, we found that vCA1-BA and vCA1-LHA shock responsive neurons were not biased to respond to all modalities of salient stimuli, as both projection populations responded to tones at chance levels (Supplementary Fig. 1E, F). These data indicate that a population of vCA1-BA neurons is specialized to exhibit reliable representations of aversive shocks.

### vCA1 correlated activity is proportional to memory strength.
Considering the enrichment of shock-responsive neurons within the vCA1-BA projection, we next identified vCA1 shock-responsive cells in a whole-population imaging paradigm (Fig. 2a) and examined whether these cells may be used to recruit relevant ensembles of neurons during context retrieval. Recent work utilizing in vivo recordings have documented changes in correlated activity within the HPC after contextual memory formation[25–29]. Still, how correlated activity is related to memory strength, and whether these correlated networks emerge within the vCA1-BA population to mediate memory retrieval is unclear. In order to record from large populations of vCA1 neurons simultaneously, we expressed GCaMP6f under the Synapsin promoter and tracked the activity of the same vCA1 neurons during the 3-day contextual fear conditioning paradigm as described above (Fig. 2a). We first assessed changes in correlated activity between individual vCA1 neurons within the same FOV in mice with a sufficiently large number of cells/FOV (Supplementary Fig. 2I) by comparing the synchrony of $Ca^{2+}$ transients between neuron pairs across context conditioning days. Significantly correlated pairs of neurons were defined based on the correlation coefficients of $Ca^{2+}$ transient events within one second time bins during context exploration as previously described[28,30,31] (Fig. 2b, Supplementary Fig. 2A–C). We then utilized these correlation coefficients to conduct graph theory network analysis which can be useful for identifying functional connectivity features[31,32], which may be essential for executing complex cognitive functions such as associative memory retrieval. Network correlation graphs were generated for each FOV across days by drawing a connection between all correlated pairs (Fig. 2b), and graph theory analysis metrics were then computed in order to evaluate the participation of individual neurons in higher-orders of correlated network activity. We focused our analysis on three graph theory metrics, including the number of correlated pairs per neuron, membership of a neuron within a larger inter-connected group (component probability), and the extent to which a given neuron's pairs were also correlated with each other (clustering coefficient). Analysis of these features would allow us to identify highly interconnected nodes that may enhance network communication (as measured by number of correlated pairs per neuron) and distinct network communities that may restrict the spread of neuronal signals within functional modules (as measured by component probability and clustering coefficient)[31,32]. Utilizing these metrics, we found a robust increase in vCA1 correlated activity during context retrieval in fear-conditioned mice, but not neutral context exposed mice that did not receive a shock during context encoding (Supplementary Fig. 3). This change in correlated activity was characterized by an increase in the number of coincident $Ca^{2+}$ transients between cell pairs (Supplementary Fig. 2D), it was most pronounced during non-freezing bouts (Supplementary Fig. 2E), and could not be explained by the change in $Ca^{2+}$ transient rate across days (Supplementary Fig. 2F–H). Moreover, this effect was not due to shock treatment alone, as vCA1 fear-conditioned mice did not display an increase in correlated activity when exposed to a novel context B on day 3 (Supplementary Fig. 3H–J) or day 2 (Supplementary Fig. 7A–E) following shock treatment.

We next identified vCA1 shock-responsive neurons during context encoding (Supplementary Fig. 4A) and assessed whether this population was specialized to participate in the correlated network ensemble during retrieval. Similar to our projection-specific imaging dataset, vCA1 shock cells identified in our

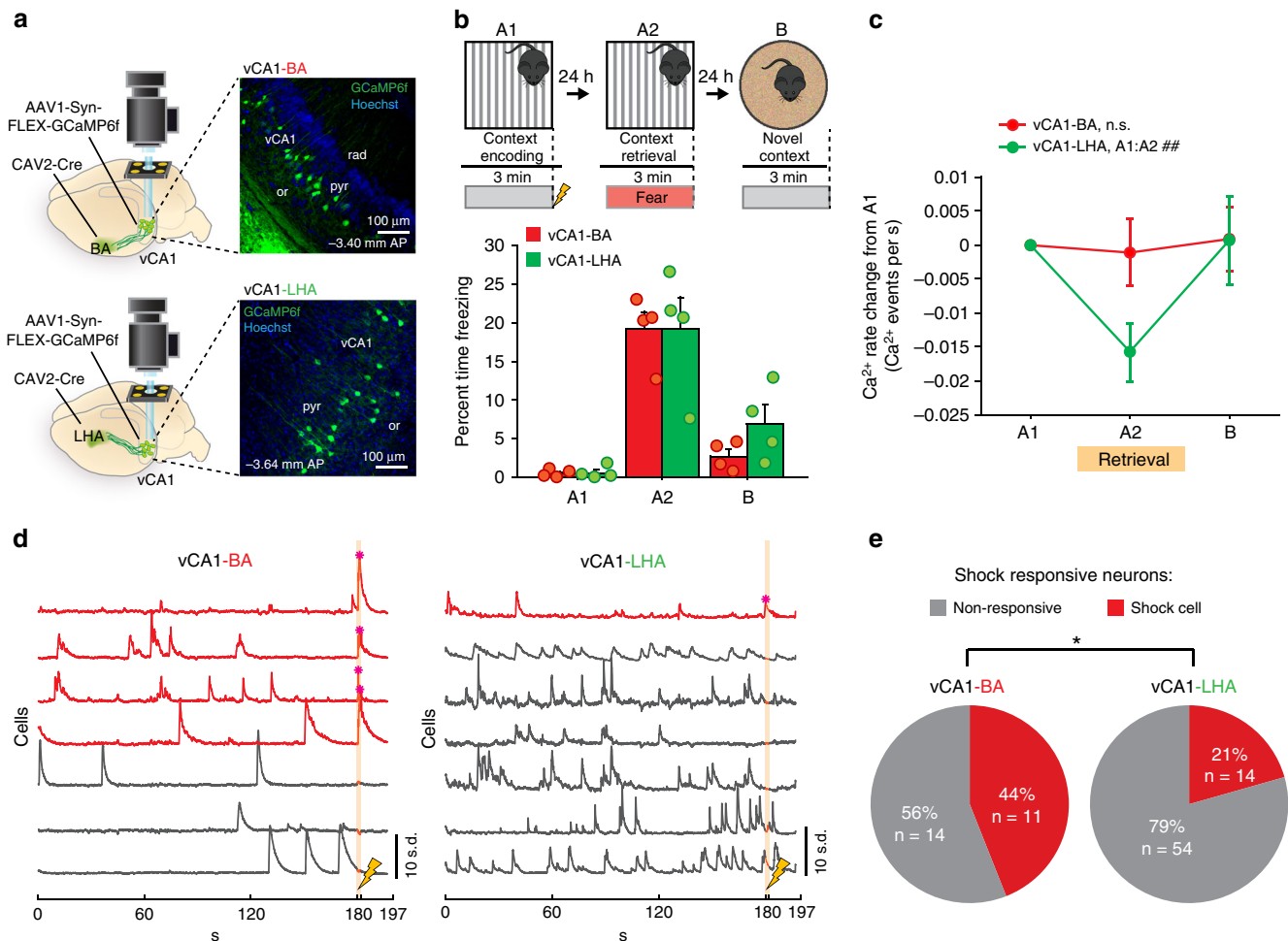

**Fig. 1 The vCA1-BA projection is enriched in shock responsive neurons. a** Experimental design for vCA1 projection-specific Ca$^{2+}$ imaging. Cre-dependent GCaMP6f was virally expressed in vCA1 and CAV2-Cre injected in either the BA (top) or LHA (bottom) in order to selectively express GCaMP6f in vCA1-BA or vCA1-LHA projecting neurons. The main effects of this experiment were consistent across all mice within each group. **b** vCA1-BA and vCA1-LHA mice exhibited similar freezing across contextual fear conditioning Ca$^{2+}$ imaging days (repeated-measures ANOVA; % freezing*group interaction, $F_{(1,6)} = 0.46$, $p = 0.48$, $N_{BA} = 4$, $N_{LHA} = 4$). **c** The vCA1-LHA (but not vCA1-BA) projection has a significant decrease in Ca$^{2+}$ event rate during A2 retrieval (BA $N_{cells} = 25$, LHA $N_{cells} = 68$|Mann–Whitney between groups with Bonferroni alpha correction; A2–A1 $U = 681.00$, $p = 0.14$, B–A1 $U = 798.50$, $p = 0.66$|Friedman across 3 days (within-group) with post-hoc wilcoxon sign rank A1:A2 and B:A1 (with Bonferroni alpha correction); BA: Friedman $X^2(2) = 2.06$, $p = 0.36$, A1:A2 $Z = -1.26$, $p = 0.36$; LHA: Friedman $X^2(2) = 9.93$, $p = 0.0070$, A1:A2 $Z = -3.34$, ##$p = 0.0008$, B:A1 $Z = -0.45$, $p = 0.65$). **d** Example Ca$^{2+}$ traces from a vCA1-BA (left) and vCA1-LHA (right) FOV during A1 context encoding showing shock responsive neurons in red and Ca$^{2+}$ transients during the shock marked with an asterisk. **e** The vCA1-BA projection has significantly more shock cells than the vCA1-LHA projection (Chi-squared test of proportions $X^2(1) = 5.10$, *$p = 0.0240$, $N_{BA} = 25$, $N_{LHA} = 68$). Error bars, ±s.e.m. Statistical tests comparing distributions were two-sided. Source data are provided as a Source Data file.

whole-population imaging paradigm were not responsive to salient tones (Supplementary Fig. 7F–I). We found that during context retrieval, vCA1 shock cells were not biased to be correlated with each other, but rather, the majority of vCA1 shock cells became correlated with non-shock cells (Fig. 2c, Supplementary Fig. 4B). We next compared cells that were correlated with shock cells during context retrieval (A2 shock-partner) to those that were not (A2 non-shock-partner), and tracked their activity across context exposures (Fig. 2d, Supplementary Fig. 8). We first examined the spatial distribution of vCA1 correlated and non-correlated cell pairs within these three subpopulations, and found that vCA1-correlated neuron pairs were spatially closer than non-correlated cell pairs, and this effect did not differ across context conditioning day or cell type (Supplementary Fig. 5A–C). Moreover, all three-cell populations underwent a significant degree of reorganization of their

correlated pair partners across days, as only ~10% of A1 correlated pairs persisted in A2 or B (Supplementary Fig. 5D, left panel).

We next assessed the magnitude of correlated activity within these three-cell populations across context conditioning days, and found that A2 shock-partners accounted for most of the change in vCA1 correlated activity during context retrieval, as this population exhibited the largest increase in correlated pair ratio, component probability, and clustering coefficient during context retrieval, while the non-partner population did not change across days (Fig. 2e–g, Supplementary Fig. 4E). This effect was not due to differences in the level of activity between A2 shock-partner and A2 non-shock-partner populations, as they exhibited similar Ca$^{2+}$ event rates across conditioning days and between A2 non-freeze and freeze bouts (Supplementary Fig. 4C, D), and the correlated pair ratio was not correlated with the A2 non-freeze

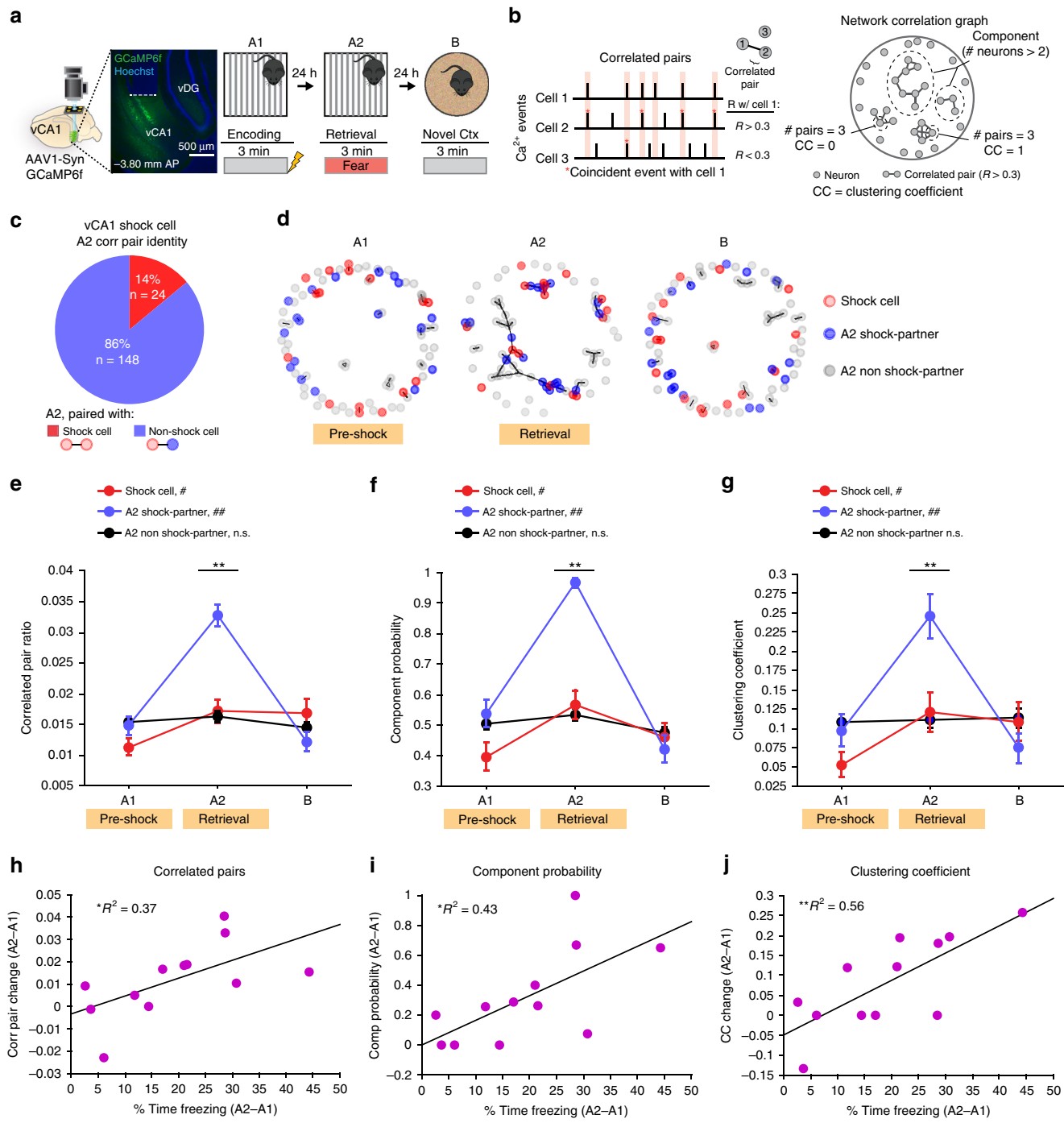

Ca$^{2+}$ activity rate for any vCA1 cell population (Supplementary Fig. 6A). Moreover, applying a correlated pair significance $R$ threshold based on shuffled cell activity (which would control for correlation differences due to the magnitude of Ca$^{2+}$ activity alone) recapitulated the effects observed at the $R \geq 0.3$ significance threshold (Supplementary Fig. 6B–D). This effect was also not driven by a sampling bias of A2 shock-partners (which by definition have at least 1 correlated pair in A2), as this effect was not seen in randomly sampled A2 non-shock-partners with at least 1 correlated pair in A2 (Supplementary Fig. 4F). Finally, the observed increase in correlated activity during context retrieval in A2 shock-partner cells did not reflect a general cellular property of these neurons to exhibit increased correlated activity in states

of high network coherence, as this specialization of increased correlated activity was unique to the conditioned context A2 (Supplementary Fig. 5D, right panel). Thus, A2 shock-partner cells may serve as highly interconnected nodes that form a distinct network community with vCA1 shock cells during contextual fear memory retrieval.

Given that vCA1 shock cells and their A2 partners accounted for most of the change in vCA1-correlated activity during context retrieval, we next assessed if the magnitude of correlation change within this sub-population was related to the strength of memory retrieval (as defined by change in time freezing, A2–A1). We found that across all metrics, the increase in correlated activity of vCA1 shock cells and their A2 partners was positively

**Fig. 2 Correlated activity with vCA1 shock cells is proportional to memory strength. a** Experimental design for vCA1 Ca$^{2+}$ imaging in contextual conditioning. GCaMP6f was virally expressed and a GRIN lens implanted to target the vCA1 pyramidal layer (left). Ca$^{2+}$ activity was imaged across days; Mice received a footshock at the end of A1 (right). The main effect of increased correlated activity in vCA1 during memory retrieval was replicated in five independent datasets. **b** Correlation graph analysis design; Top: mock traces of Ca$^{2+}$ events demonstrating calculation of significantly correlated pairs (Pearson's $R > 0.3$, cell 1 correlated with cell 2, but not cell 3). Bottom: Example of a correlation graph with extrapolated graph parameters constructed using Pearson's $R$ correlation matrix. Cell pairs with significant correlation coefficient are connected by a line. **c** Proportion of vCA1 shock cell correlated pairs with other shock cells (red) or non-shock cells (blue). **d** Representative correlation graphs with vCA1 shock cells colored in red, and their correlated pairs during context retrieval (A2 shock-partners) colored in blue across conditioning days. **e–g** Graph parameters of vCA1 shock cells (red), A2 shock-partners (blue), and all other cells (A2 non-shock-partner, black) across conditioning days ($N_{shock} = 111$, $N_{A2\ shock-partner} = 121$, $N_{A2\ non-shock-partner} = 616$; Kruskal–Wallis between groups with Bonferroni alpha correction; corr pair ratio: A2–A1 **$p < 0.0001$, $H(2) = 49.65$, B–A1 $p = 0.08$, $H(2) = 5.01$; comp memb: A2–A1 **$p < 0.0001$ $H(2) = 33.06$, B–A1 $p = 0.18$ $H(2) = 3.47$; CC: A2–A1 **$p < 0.0001$, $H(2) = 26.24$, B–A1 $p = 0.65$ $H(2) = 0.87$ | Wilcoxon sign rank A1, A2; corr pair ratio: shock $Z = -2.47$ #$p = 0.0134$, A2 shock-partner $Z = -6.58$ ##$p < 0.0001$, A2 non-shock-partner $Z = -0.61$, $p = 0.54$; comp memb: shock $Z = -2.52$ #$p = 0.0116$, A2 shock-partner $Z = -6.04$, ##$p < 0.0001$, A2 non-shock-partner $Z = -1.01$ $p = 0.31$; CC: shock $Z = -2.21$, #$p = 0.0270$, A2 shock-partner $Z = -4.31$ ##$p < 0.0001$, A2 non-shock-partner $Z = -0.32$, $p = 0.75$). **h–j** Linear regression; mean change in correlation graph parameters from context encoding to retrieval from vCA1 shock cells and their A2 shock-partners per FOV (A2–A1, $N_{mice} = 12$) versus memory retrieval strength (change in % time freezing A2–A1) (# corr pair: $F_{(1,10)} = 5.95$, *$p = 0.0349$, $R^2 = 0.37$; comp memb: $F_{(1,10)} = 7.56$, *$p = 0.0205$, $R^2 = 0.43$; CC: $F_{(1,10)} = 12.57$, **$p = 0.0053$, $R^2 = 0.56$). Error bars, ±s.e.m. Statistical tests comparing distributions were two-sided. Source data are provided as a Source Data file.

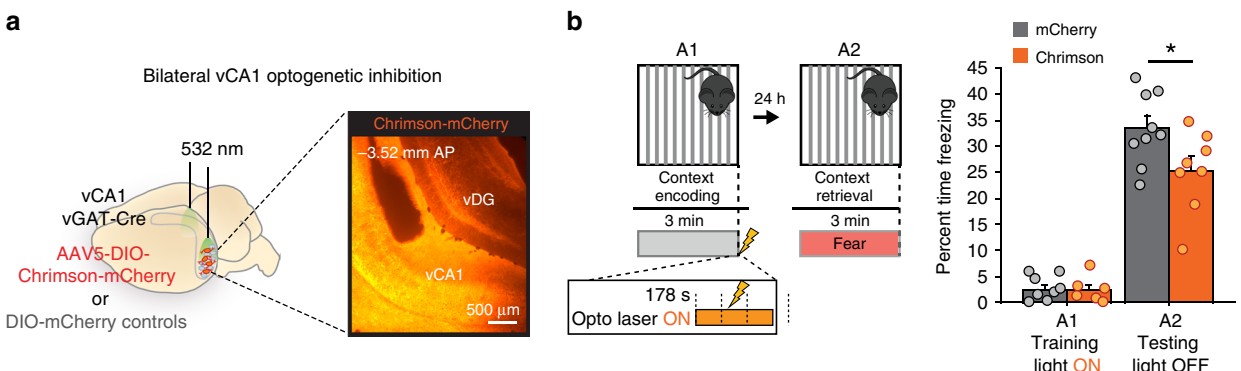

**Fig. 3 vCA1 shock activity is necessary for contextual memory encoding. a** Experimental design, bilateral vCA1 optogenetic activation of vGAT-Chrimson cells with 532 nm laser. Left: DIO-Chrimson or DIO-mCherry was injected into vCA1 of vGAT-Cre mice bilaterally and fiber optics implanted. Right: Example image of vCA1 with expression of Chrimson, and fiber optic targeting. The main effect of this experiment was replicated in an independent dataset within the lab. **b** Left: Experimental design, laser stimulation was delivered for 6 s during the footshock in A1 context encoding (ON 178–184 s). Right: Bilateral vCA1 shock silencing with vGAT-Chrimson stimulation during context encoding significantly reduced % time freezing during context retrieval relative to vGAT-mCherry control mice (repeated-measures ANOVA; % time freezing*genotype interaction $F_{(1,15)} = 6.26$, *$p = 0.0244$, Chrimson $N = 8$, mCherry $N = 9$). Error bars, ±s.e.m. Statistical tests comparing distributions were two-sided. Source data are provided as a Source Data file.

correlated with the strength of memory retrieval across animals (Fig. 2h–j, Supplementary Fig. 8). These results suggest that correlated activity with vCA1 shock-responsive neurons may be causally related to contextual fear memory retrieval.

**vCA1 shock response is necessary for memory encoding.** Considering that the strength of memory retrieval was proportional to the magnitude of correlation change within the network of vCA1 shock cells and their A2 partners, we next assessed whether the vCA1 shock response in context encoding was necessary for the formation of a contextual fear memory. To assess this, we virally expressed a Cre-dependent red-shifted opsin Chrimson or mCherry no-opsin control virus within vCA1 of vGAT-Cre mice bilaterally and implanted fiber optics to deliver laser stimulation during behavior (Fig. 3a). We then exposed Chrimson and mCherry mice to the conditioning context A1 in the same manner as our imaging paradigm described above, and delivered 6-s of laser stimulation during the shock period only (178–184 s) to selectively disrupt vCA1 activity during the shock period and assessed the effect on

freezing behavior during context retrieval A2 (Fig. 3b, left panel). We found that vGAT-Chrimson mice exhibited significantly less freezing behavior during A2 retrieval relative to mCherry controls (Fig. 3b, right panel). These data suggest that the vCA1 shock response during context encoding is necessary for the formation of contextual fear memory.

**vCA1 shock response is necessary for correlated activity.** We next investigated whether the vCA1 shock response during context encoding was necessary for the emergence of correlated activity during retrieval, or if these correlated networks would form independently of aversive shock representations at the level of vCA1. To accomplish this, we employed a simultaneous Ca$^{2+}$ imaging and unilateral optogenetic silencing manipulation to selectively inhibit vCA1 activity during the shock period in context encoding, and assessed the effect on correlated activity during retrieval (Fig. 4c). Additionally, this unilateral approach would allow us to dissociate effects of circuit activity manipulations from group differences in freezing behavior, as prior studies have demonstrated that unilateral HPC manipulations are

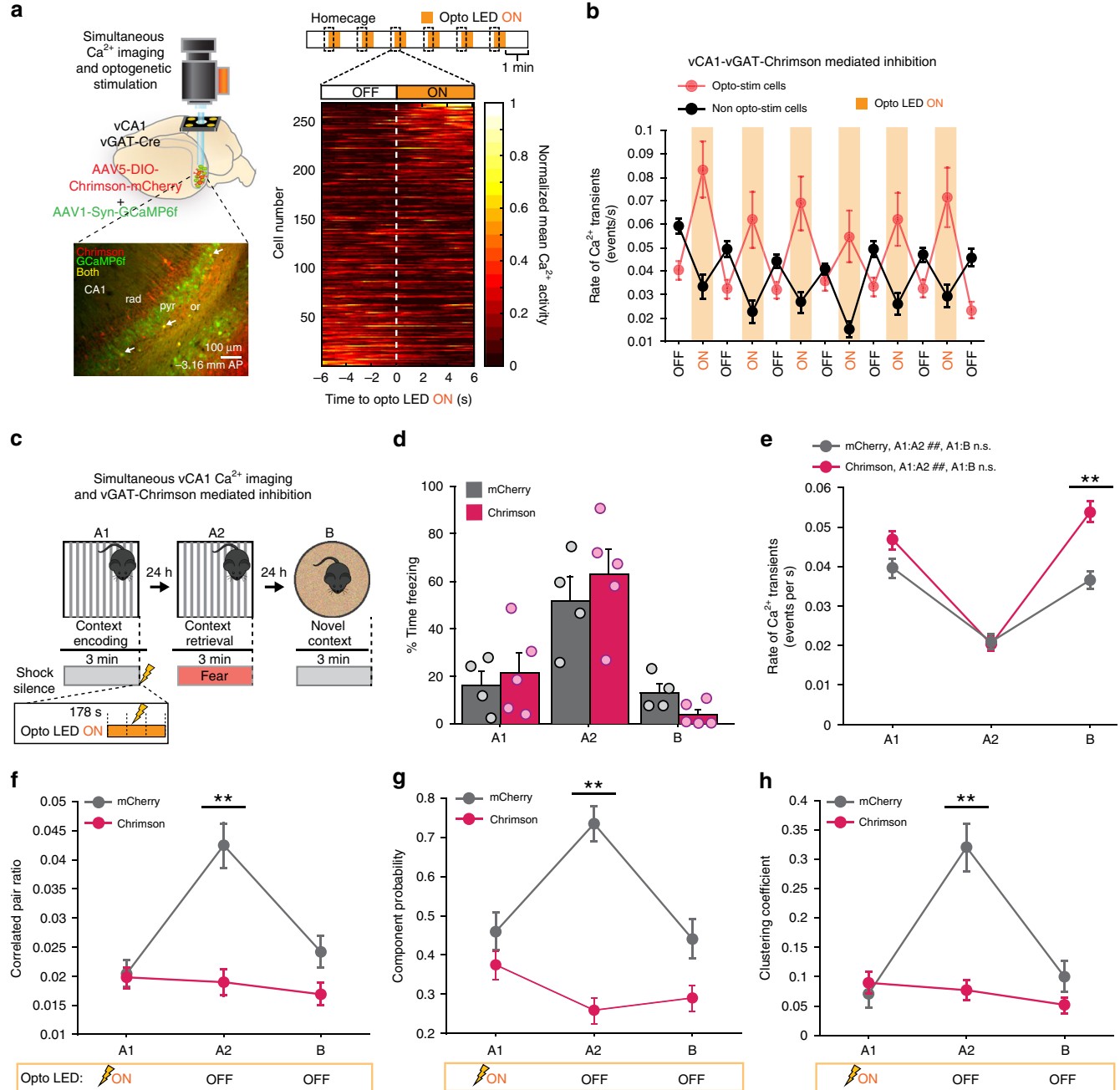

**Fig. 4 vCA1 shock activity is necessary for correlated activity during memory retrieval. a** Experimental design, simultaneous vCA1 Ca²⁺ imaging and unilateral optogenetics. Top left: DIO-Chrimson and Syn-GCaMP6f were co-injected into vCA1 in vGAT-Cre mice to selectively express Chrimson in vGAT interneurons and GCaMP6f in all neurons. GCaMP6f was excited by a blue LED (455 nm). Chrimson was excited by an amber LED (590 nm). Bottom left: Representative image of vCA1 with co-expression of Chrimson (red), GCaMP6f (green), and cells expressing both (yellow) indicated by white arrow. Right: Heat map of Ca²⁺ activity 6 s before and after turning on the optoLED (average of six pulses in homecage, cells sorted by magnitude of change ON–OFF). The main effect of this experiment was replicated in an independent dataset within the lab. **b** Rate of Ca²⁺ transients in homecage during optoLED 6-s pulses ON in opto-stim (red) and non-opto-stim (black) cell populations demonstrating optogenetic silencing of vCA1 using vGAT-Chrimson activation. **c** Experimental design, mice were imaged in a 3-day contextual fear paradigm, with the optoLED turned ON during the footshock in A1 (ON from 178 to 184 s) to selectively disrupt the vCA1 shock response. **d** Unilateral shock silencing in A1 did not disrupt freezing behavior during retrieval (repeated-measures ANOVA; % time freezing*genotype interaction $F_{(1,7)} = 1.22$, $p = 0.33$, $N_{mCherry} = 4$, $N_{Chrimson} = 5$). **e** vCA1-mCherry and vCA1-Chrimson neurons exhibited a similar decrease in Ca²⁺ event rate during context retrieval (mCherry $N_{cells} = 102$, Chrimson $N_{cells} = 190$; Mann–Whitney between groups with Bonferroni corrected alpha: A1 $U = 8636.00$, $p = 0.13$, A2 $U = 8907.00$, $p = 0.26$, B $U = 7071.00$, **$p = 0.0001$| Wilcoxon sign rank with Bonferroni corrected alpha; mCherry: A1:A2 $Z = -6.16$ ##$p < 0.0001$, A1:B $Z = -1.329$ $p = 0.18$; Chrimson: A1:A2 $Z = -9.13$ ##$p < 0.0001$, B:A1 $Z = -2.16$ $p = 0.0311$). **f–h** Disrupting the vCA1 shock response during encoding impaired the formation of correlated activity during retrieval across all graph parameters in vGAT-Chrimson mice (mCherry $N_{cells} = 102$, Chrimson $N_{cells} = 190$; Mann–Whitney between groups with Bonferroni corrected alpha; corr pair ratio A2-A1 $U = 6713.00$, **$p < 0.0001$, B-A1 $U = 8399.00$, $p = 0.06$; comp probability A2-A1 $U = 6776.00$, **$p < 0.0001$, B-A1 $U = 9217.00$, $p = 0.49$; CC A2-A1 $U = 6376.00$, **$p < 0.0001$, B-A1 $U = 9025.50$, $p = 0.33$). Error bars, ±s.e.m. Statistical tests comparing distributions were two-sided. Source data are provided as a Source Data file.

insufficient to disrupt memory encoding and freezing behavior during retrieval in contextual memory tasks[33,34]. We used a Cre-dependent virus to express the excitatory opsin Chrimson in inhibitory interneurons in vGAT-Cre transgenic mice, while simultaneously expressing GCaMP6f in all cells (Fig. 4a, Supplementary Fig. 9A). We found that stimulation of vGAT-Chrimson+ neurons led to robust and reliable silencing in non-optogenetically modulated cells (non-opto-stim cells) (Fig. 4a, b, Supplementary Fig. 9B, C). We then used this approach to selectively silence vCA1 activity during the shock period in vGAT-Chrimson mice, and assessed the effect on correlated activity during retrieval in these mice relative to control vGAT-mCherry mice (Fig. 4c). As expected after a unilateral vCA1 manipulation in a contextual memory task, freezing behavior was similar between vGAT-mCherry and vGAT-Chrimson mice across days (Fig. 4d). In addition, both groups exhibited a similar decrease in vCA1 $Ca^{2+}$ event rate in A2 (Fig. 4e). However, despite these similarities in freezing behavior and $Ca^{2+}$ event rate, optoLED light delivery during the shock period in context encoding abolished the increase in correlated activity during context retrieval in vGAT-Chrimson, but not vGAT-mCherry mice (Fig. 4f–h, Supplementary Fig. 9D). This effect of vGAT-Chrimson silencing during context encoding was specific to the shock period, as silencing vGAT-Chrimson neurons for 6 s outside of the shock period did not disrupt the formation of correlated activity during context retrieval (Supplementary Fig. 10). Moreover, this effect was not due to an unintentional disruption of vCA1 activity in vGAT-Chrimson+ mice by the blue GCaMP LED during opto light-off conditions (considering Chrimson's broad excitation spectrum), as these mice were able to form correlated activity during a subsequent tone-retrieval paradigm in which mice were trained with tone–shock pairings without opto stimulation (Supplementary Fig. 11). Thus, the increased vCA1-correlated activity we observe during memory retrieval cannot be explained by freezing behavior and decreased $Ca^{2+}$ event rate, but rather requires a representation of the aversive shock at the level of vCA1 during memory encoding. These results suggest that contextual fear memory engrams in vCA1 are composed of ensembles of neurons that are active during encoding and become correlated with valence encoding neurons (such as shock cells) during retrieval.

## Discussion

While decades of work have established a role for the HPC in contextual memory based on static markers of prior cell activity[2–4,6], the dynamic signatures of cell activity by which the HPC encodes and retrieves these contextual memories is not well understood. Both experimental data[25–27,35] and computational models[36–38] have proposed that synchronous activity within the HPC may facilitate long-term synaptic plasticity for memory formation and retrieval. Our work identifies a putative signature of memory retrieval within synchronous ensembles of neurons in vCA1 which is related to memory strength, and incorporates neurons responding to the unconditioned stimulus during training. Moreover, we demonstrate that the vCA1 shock response during context encoding is necessary for the emergence of these functional correlation networks during memory retrieval. This finding may be explained by recent studies which have found that increased neuronal excitability promotes cell recruitment into a memory trace or engram[1,39], as neuronal excitability enhances synaptic plasticity and promotes synaptic strengthening[38]. Therefore, by extension, the shock may result in increased recruitment of vCA1-BA projecting neurons (as this projection is enriched in shock-responsive neurons) to the contextual fear memory engram by increasing neuronal excitability in vCA1 at the time of associative memory encoding. Importantly, the correlations described here are captured over slow time scales (within 1 s time bins), which may be mediated by subthreshold activity or $Ca^{2+}$ plateau potentials which have recently been described in a CA1 synaptic plasticity rule which operates over the timescale of seconds[40]. Finally, when considering why correlation with vCA1 shock-responsive neurons may promote memory retrieval, we propose two possible mechanisms. First, considering that only ~50% of vCA1-BA projecting neurons are responsive to aversive shocks, vCA1 shock cells that project to the amygdala may form a functional correlation network with other vCA1-BA projecting neurons that are not responsive to the aversive US, but rather encode a representation of the conditioned context. In this model, correlated activity between these projecting neurons could lead to enhanced communication between the HPC-amygdala during contextual fear memory retrieval by increasing synchronized synaptic release within the BA, which would then engage downstream circuits to promote freezing behavior (Fig. 5a). Alternatively, vCA1 shock cells could form a functional correlation network with neurons that encode the context representation but that do not project to the BA, which could promote long-range synchrony and enhance communication across multiple brain regions that project to the amygdala (Fig. 5b). This could be accomplished for example through vCA1 shock cell synchrony with vCA1 neurons projecting to the medial prefrontal cortex (mPFC in Fig. 5b), whose inputs to the BA are known to impact fear expression[41]. Still, in the current study, the participation of vCA1-BA shock cells in functional correlation networks during contextual fear memory retrieval were not examined directly due to the limitations of the retrograde GCaMP-labeling techniques employed for imaging $Ca^{2+}$ activity of projection-specific populations. Therefore, future studies examining the activity properties of projection-specific vCA1 populations within the greater vCA1 population are required to support these putative models of vCA1-BA shock cell-mediated memory retrieval. Taken together, our results identify a putative circuit mechanism by which vCA1 may encode and retrieve emotional contextual memories across ensembles of co-active neurons. Such patterns of synchronous activity have recently been proposed to underlie the persistence of memories over long periods of time[25]. A dissection of the mechanisms underlying the emergence of these correlations during memory formation (such as the recruitment of local interneurons) will therefore be critical for our ability to understand and improve long-term memory.

## Methods

**Animal subjects.** Procedures were conducted in accordance with the U.S. NIH Guide for the Care and Use of Laboratory Animals and the New York State Psychiatric Institute Institutional Animal Care and Use Committees at Columbia University. Adult male C57BL/6J mice were supplied by the Jackson Laboratory, and vGAT-IRES-Cre mice[42] were bred on a C57BL/6J background. Mice were used for experiments at 8 weeks of age and provided with unrestricted access to food and water. All mice were housed 2–5 per cage on a 12 h light/dark schedule with lights off at 6:00 p.m., with ambient temperature ~74 °F and ~20% humidity. Experiments were conducted during the light portion.

**Viral constructs.** Adeno-associated viruses for optogenetic manipulations were packaged and supplied by the UNC Vector Core Facility at titers of ~4–8 × $10^{12}$ vg/ml (AAV5-Syn-Flex-ChrimsonR-tdTomato; AAV5-Syn-DIO-mCherry). For calcium imaging, viruses (AAV1-Syn-GCaMP6f.WPRE.SV40; AAV1-Syn-Flex-GCaMP6f. WPRE.SV40) were packaged and supplied by UPenn Vector Core at titers ~6 × $10^{12}$ vg/ml and viral aliquots were diluted prior to use with artificial cortex buffer to ~2 × $10^{12}$ vg/ml. The CAV2-Cre virus was packaged and supplied by the Larry S Zweifel laboratory in collaboration.

**Stereotactic surgeries.** For all surgical procedures, mice were anesthetized (1.5% isoflurane,1 L/min $O_2$) and head-fixed in a stereotax (David Kopf, Tujunga, CA). Ophthalmic ointment was applied for eye lubrication, fur was shaved and incision

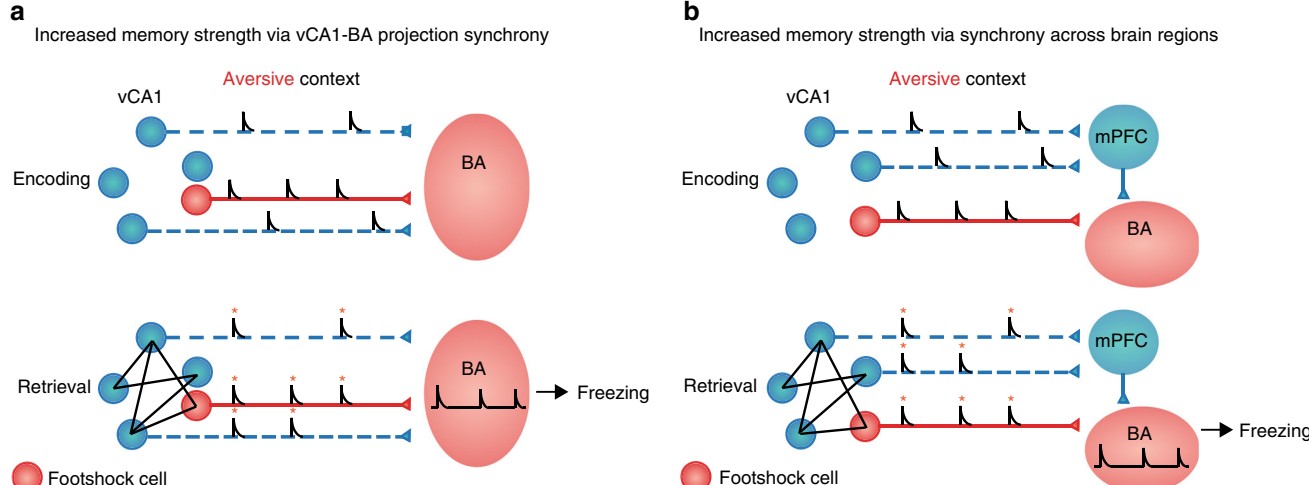

**Fig. 5 Putative mechanisms for vCA1 shock cell-dependent memory retrieval.** Putative mechanism for vCA1 shock cell correlation-mediated memory retrieval. **a** vCA1 shock cells (red) and their correlated A2 partners (blue) represent vCA1-BA projecting neurons, and their synchronized $Ca^{2+}$ transients during retrieval (bottom) results in coincident synaptic release within the BA to facilitate fear expression and induce freezing behavior. **b** vCA1 shock cells (red) represent vCA1-BA projecting neurons and their correlated A2 partners represent a separate vCA1 projection with indirect projections to the amygdala, such as the vCA1-mPFC projection (blue). Synchronized $Ca^{2+}$ transients between vCA1-BA (shock cells) and vCA1-mPFC projections results in coincident synaptic release within the amygdala to facilitate fear expression and induce freezing behavior.

site sterilized, and body temperature maintained with a T/pump warm water recirculator (Stryker, Kalamazoo, MI). Subcutaneous saline and carpofen were provided peri-operatively and post-operatively for 3 additional days for analgesia and hydration.

For in vivo $Ca^{2+}$ imaging surgical procedures[9,43], three skull screws (FST, Foster City, CA) were inserted around the implantation site, a craniotomy was made using a dental drill and dura removed from the brain surface with fine tweezers and a stream of sterile saline and absorptive spears (Fine Science Tools (FST), Foster City, CA). Mice were then injected unilaterally (in mm, from brain tissue at site, vCA1: −3.16 AP, 3.25 ML, −3.85, −3.50, −3.25 DV) with 500 nl of GCaMP6f with a Nanoject syringe (Drummond Scientific, Broomall, PA) followed by implantation of a ~0.5 mm diameter, ~6.1 mm long GRIN lens (Inscopix, Palo Alto, CA) over the injection site. The lens was slowly lowered in 0.1 mm DV steps to the target depth (in mm, from skull at craniotomy, vCA1: −3.16 AP, 3.50 ML, −3.50 DV) and then fixed to the skull with dental cement (Dentsply Sinora, Philadelphia, PA). At the completion of surgery, the lens was protected with a cap of lens paper and liquid mold rubber (Smooth-On, Lower Macungie, PA), and imaging experiments were started 3 weeks later. For CAV-Cre retrograde surgeries, CAV-Cre virus was injected at the following coordinates (~450 nl virus per AP site; LH: −2.0 and −2.30 AP, 0.75 ML, −5.25, −5.0, −4.75 DV; BA: −1.70 and −2.00 AP, 3.0 ML, −4.25, −4.0 DV); For vGAT-Cre surgeries (on a pure 129 background strain) the following ML coordinates were used in order to target vCA1: viral injection site ML 3.75 mm, lens implantation site ML 3.90 mm). For simultaneous vCA1 optogenetic and $Ca^{2+}$ imaging, GCaMP6f and opsin virus were pre-mixed (2:1, opsin: undiluted-GCaMP virus) prior to injection in brain tissue.

For bilateral optogenetic silencing experiments (also conducted in pure 129 vGAT-Cre background strain), undiluted opsin virus was injected at the following coordinates (5 pulses of 32.1 nl volume per DV site; vCA1: −3.16 AP, 3.70 ML, −3.85, −3.50, −3.25 DV) and fiber optics were implanted at the following coordinates (vCA1: −3.16 AP, 3.85 ML, −3.50 DV).

**Contextual fear conditioning.** For CFC imaging, mice were run through a 3-day contextual fear conditioning paradigm. On day 1, mice were placed in a standard fear conditioning shock box (Med Associates, ENV-010MD) with the following contextual cues: *Context A*: anise scent, white noise, floor shock bars exposed, shock box doors closed, and a light on within the chamber, and were allowed to explore the context for 3 min prior to receiving a 2 s 0.7 mA strength foot shock. On day 2, mice were placed back into the same context for 3 min to assess for freezing during CFC retrieval. On day 3, mice were placed in a modified fear conditioning box for 3 min in the same room but with the following modified contextual cues: *Context B*: lemon scent, no white noise, floor shock bars covered with an insert and with bedding, shock box doors open so that mice could see distal cues within the behavior room, and chamber light off. For tone–shock conditioning, at the end of the 3 min of day 2 retrieval in Context A, mice were treated with 3 tone–shock pairings, with a 2 s foot shock delivered in the last 2 s of a 20 s, 2 kHz, 90 db tone. On day 3 in Context B after 3 min of novel context exploration, the 20 s tones were presented to mice to assess for tone-retrieval (tones starting at minutes 3, 4, and 5). Behavior was recorded with EthovisionXT 11 video software,

and freezing was hand-scored by a blinded experimenter using ObserverXT 12 scoring software (Noldus, Leesburg, VA).

**Freely moving $Ca^{2+}$ imaging.** Three weeks after surgery, mice were head-fixed on a stereotactic frame while under 1.5% isoflurane with 1 L/min oxygen flow in order to check GRIN lenses for GCaMP expression with a miniaturized microscope (Inscopix, Palo Alto, CA)[9,43]. The protective rubber mold was removed from the lens and lens cleaned with ethanol wipes and lens paper. A magnetic baseplate was screwed onto the miniscope and lowered over the implanted GRIN lens until the FOV came into focus. The baseplate was dental cemented in place onto the mouse headcap if GCaMP+ neurons were visualized. After drying the baseplate was unscrewed from the miniscope and miniscope was removed from the animal's headcap, and baseplate was covered with a customized lens cover (Inscopix, Palo Alto, CA). Once baseplated, the same microscope was used for every imaging session with that mouse and the focal plane of the microscope was left unchanged in order to ensure a constant FOV across imaging sessions. Imaging sessions were started the following day. Prior to each imaging session, mice were briefly anesthetized (<5 min) in order to attach the miniscope to the baseplate, and a snap shot of the FOV was taken and matched between days to ensure exact placement of the miniscope within the baseplate in order to maintain a constant FOV across days. All mice were allowed to recover from anesthesia for 30 min before beginning imaging, and $Ca^{2+}$ videos were recorded with nVista and nVoke acquisition software (Inscopix, Palo Alto, CA), and behavior and $Ca^{2+}$-imaging video acquisition were synchronized with a TTL-triggering system (a TTL pulse from Etho-Vision XT 10 and Noldus IO box system was received by nVista and nVoke DAQ boxes to commence imaging sessions). $Ca^{2+}$ videos were acquired at 15 or 20 frames per second with 66.56 ms exposure, and the same LED power settings were used for each mouse throughout the series of imaging sessions.

**Bilateral optogenetic stimulation.** Four-week after viral injection and fiber-optic implantation, mice were handled and habituated to fiber-optic adapter cables for 2 days prior to beginning behavioral experiments. For vGAT-Chrimson stimulation, ~5–8 M were delivered via a 523 nm 100 mW laser (Opto Engine, Midvale, UT) to fiber optics implanted in mouse brain using a fiber-optic patch cable[10]. EthoVision XT 11 software and a Noldus IO box system were used to record live-tracking of mice while they explored the CFC box and to trigger the laser ON during the specified time period.

**Simultaneous $Ca^{2+}$ imaging and optogenetic stimulation.** Simultaneous calcium imaging and optogenetic manipulation was performed using the nVoke miniature microendoscope from Inscopix. Briefly, the nVoke system uses two LEDs (455 nm blue LED, 590 nm amber LED) with different wavelengths to simultaneously excite GCaMP for imaging and activate a red-shifted opsin (in this case Chrimson-mCherry) for optogenetic manipulation. Custom code written through Med-Associates software was used to control the fear conditioning boxes and TTL connection with the Inscopix system allowing for precise timing of LED and shock delivery.

For nVoke imaging, a 3-day CFC paradigm was performed as described above. On the first day of CFC, during a period extending from 2 s before the shock to 2 s after the shock (total 6 s) continuous illumination with the 590 nm LED was delivered unilaterally to vCA1 through the implanted GRIN lens. On days 2 and 3 of the CFC paradigm no amber LED stimulation was given. Immediately after the 3rd day of CFC mice were returned to their homecages and received alternating ON–OFF epochs of amber LED stimulation while simultaneously imaging GCaMP activity. Six epochs (each 30 s) of amber LED stimulation were interspersed with seven epochs (each 60 s) without amber LED stimulation.

**Histology and confocal and epifluorescent microscopy**. For histology, mice were perfused transcardially with 4% paraformaldehyde (PFA) in 1X phosphate buffer solution (PBS), after which brains were removed and post-fixed in 4% PFA for 24 h. They were transferred to a 30% sucrose solution in 1X PBS for 2 days, after which they were flash-frozen in methylbutane and coronally sliced on a cryostat (Leica CM 3050S) at a thickness of 50 μm. Prior to mounting brain sections and cover slipping with ProLong Gold antifade reagent (Invitrogen, Carlsbad, CA), they were incubated with 1:100 Hoechst (Invitrogen, Carlsbad, CA) in 1X PBS for 10 min to label cell nuclei. No immunolabeling was applied to visualize fluorophores, as endogenous viral expression was sufficiently bright in histology preparations. After mounting, brain sections were imaged on a (Leica TCS SP8) confocal microscope using a ×10 or ×20 objective, or a (Zeiss Axiovert 200) epifluorescent microscope using a ×2.5 or ×10 objective.

In order to determine appropriate GRIN lens placements after imaging, brains were post-fixed with head-caps, lenses, and skulls in place for 1 week in 4% PFA to harden the GRIN lens tract within the brain tissue for better visualization. Brains were then transferred into 30% sucrose solution as described above, and sections were then mounted in anterior–posterior order in order to facilitate the determination of GRIN lens location for each mouse on an epifluorescent microscope.

**Image processing**. Calcium video image processing was performed using Mosaic and Inscopix Data Processing software (versions 1.0.5b and 1.2.0, Inscopix, Palo Alto, CA). Videos were temporally downsampled to 5 frames per second, spatially downsampled by a binning factor of 4, and lateral brain movement was motion corrected using the registration engine Turboreg[44,45], which utilizes a single reference frame and high-contrast features in the image to shift frames with motion to matching $XY$ positions throughout the video. Black borders from $XY$ translation in motion correction were cropped, and cell segmentation was performed using Constrained Non-negative Matrix Factorization for micro-Endoscopic data (CNMF-E)[46]. Motion corrected and cropped videos were ran through the CNMF-E algorithm with an estimated cell diameter of 18 pixels, and identified neurons were sorted by visible inspection for appropriate spatial configuration and $Ca^{2+}$ dynamics consistent with signals from individual neurons. We report the non-denoised temporal traces extracted by CNMF-E as neurons' temporal activity. These traces were z-scaled with an estimated Gaussian noise level, corresponding to a scaled version of $\Delta F/Fo$ of each neuron. $Ca^{2+}$ transient events were then defined with an event detection algorithm in which transients were $Z$-scored (mean calculated from silent time points of $Ca^{2+}$ activity with fluorescence values less than the 0.50 quantile of all fluorescence values from all cells in the FOV). $Ca^{2+}$ events were defined as transients exceeding a 2 s.d. amplitude from a 0.5 s.d. baseline, lasting a minimum duration (calculated by $[-\ln(A/Ao)/t\_half]$, where $Ao$ = amplitude at start of that transient; t_half for GCaMP6f was 200 ms, before returning to a 0.5 s.d. baseline level. Additional $Ca^{2+}$ transient rising events within detected $Ca^{2+}$ transients that were large and multi-peaked were then detected using the findpeaks function in MATLAB (Mathworks, Natick, MA) with the following parameters (MinPeakProminence = 1.5 s.d., MinPeakDistance = 1 s). All detected $Ca^{2+}$ transients were visibly inspected for each cell to verify accuracy.

In order to track the same cells across contextual conditioning days, videos from the three imaging sessions were concatenated into a single video prior to motion correction. The concatenated video was then motion corrected to a single reference frame to ensure that all frames were shifted to the same location during motion correction $XY$ translation. Motion correction accuracy was then determined by visually tracking cells across imaging days (using a hand-drawn ROI in ImageJ ROI Manager) to ensure that cell spatial locations did not shift throughout the entire concatenated video.

**$Ca^{2+}$-imaging data analysis**. $Ca^{2+}$ transient events and mouse behavior were analyzed with custom MATLAB (Mathworks, Natick, MA) functions to calculate the rate of $Ca^{2+}$ transients and correlated activity per cell while mice explored different contexts. Freezing behavior was scored using ObserverXT software, and behavior data was downsampled to five frames per second to match $Ca^{2+}$ transient data sampling. Non-parametric statistical comparisons were applied to all analyses.

**Correlated activity graph theory analysis**. For all correlated activity analysis, only mice with ≥20 cells per FOV were included in the analysis given the sampling limitation of potential correlated pair partners in sparse FOVs (see Supplementary Fig. 2I). This negated inclusion of all vCA1-BA and vCA1-LHA projection-specific imaging, as these FOVs were sparse with ~5–15 cells per mouse on average. To

assess the structure of neural correlations during vCA1 whole-population CFC imaging, a similarity matrix was computed to represent the strength of correlations among pairs of neurons within a field of view. Each neuron's temporal activity was represented as a binary time series indicating the timing of peak $Ca^{2+}$ transient amplitudes, temporally binned into one second time bins, and the Pearson's correlation coefficient was computed for each pair of neurons. An $R$ threshold of 0.3 was applied to this matrix to reduce bias from noise. Next, a graph was constructed using the NetworkX python package[47] to visualize and quantify the higher order structure of the neural correlations[31]. The correlated pair ratio (# pairs/# total cells in FOV) and clustering coefficient (# edges among neighbors/# possible edges among neighbors) were computed for each neuron. Due to the sparse nature of these neural graphs, we also determined whether each neuron was a member of a connected component containing at least two other neurons. A connected component is defined as a subgraph of nodes connected to each other by paths but not connected to any other nodes. Each neuron was assigned either a 0 or 1 indicating whether it was a component member, and the proportion of neurons in a component (component probability) was computed for each imaging session.

The mean graph parameter across cell populations were plotted for each conditioning day in the main figures, with Fig. 1 plotted as a mean change from A1 given baseline group differences between Neutral and Fear mice.

**Pearson's $R$ threshold comparison for defining significantly correlated pairs**. We compared the $R$ threshold for defining a significantly correlated cell pair based on a shuffle distribution $R$ value rather than the $R > 0.3$ threshold which has previously been described[28,30]. We shuffled the timing of calcium transients for all neurons (1000 shuffle iterations) and calculated mock Pearson's $R$ values from the shuffled calcium events to create a shuffle distribution of $R$ values for each cell pair. $R$ thresholds at $p < 0.05$ from the shuffle distribution were then applied to determine the number of correlated pairs with shuffled $R$ thresholds and compared to correlated pairs defined by $R > 0.3$ thresholds (see Supplementary Figs. 2 and 6; Supplementary Fig. 6 defined significant $R$ thresholds with an alpha that was Bonferroni corrected for the number of correlated pair comparisons made per FOV).

**Heatmap of vGAT-Chrimson $Ca^{2+}$ data**. A $Ca^{2+}$ activity heatmap during homecage optoLED pulses in vGAT-Chrimson mice was generated by taking the average calcium activity of $Ca^{2+}$ transients per cell (non-transient time bins were pre-filtered to zero baseline) per 200 ms time bin from the 6 s preceding optoLED ON and the first 6 s of each 30 s optoLED ON pulse. The calcium activity across all cells was then normalized to the max and min calcium activity time bin, and cells were sorted by their rank in average change in $Ca^{2+}$ activity from OFF to ON time bins.

**Defining shock-cells and tone-cells**. For defining cells that were significantly responsive to shocks or tones, $Ca^{2+}$ events were shuffled in time for individual cells (1000 iterations), and shock or tone rates were re-calculated from those behavioral time periods to generate a null distribution of shock or tone $Ca^{2+}$ event rates for each cell. A cell was considered a shock-cell or tone-cell if its $Ca^{2+}$ event rate during those periods exceeded a 1 SD threshold from the null distribution.

**Defining opto-stim cells**. For defining cells that were optogenetically stimulated by the optoLED during a homecage 30 s pulse session, $Ca^{2+}$ events were shuffled in time for individual cells (1000 iterations), and optoON shuffled rates were re-calculated from the first 6 s of each 30 s pulse (six pulses total) to generate a null distribution of optoON $Ca^{2+}$ event rates for each cell. A cell was considered an opto-stim cell if its $Ca^{2+}$ event rate during the optoON periods exceeded a 1 SD threshold from the null distribution. The threshold was determined by comparing $Ca^{2+}$ rates for neutral cells defined at different thresholds, and the threshold at which neutral cells showed no significant $Ca^{2+}$ event rate difference between optoON and optoOFF time periods was selected.

**Control for potential A2 shock-partner selection bias**. In order to control for a potential selection bias in our A2 shock-partner analysis (which by definition, is a cell population with at least one correlated pair in A2), we randomly sampled a matching proportion of cells from the A2 non-shock-partner population with at least one correlated pair on A2 (10,000 iterations), and created a histogram of the change in graph parameters from A2 to A1 from each randomly sampled cell population (Supplementary Fig. 4F). The true change in graph parameters and $p$-value for the A2 shock-partner population was then calculated relative to the change in these random sample distributions.

**Reporting summary**. Further information on research design is available in the Nature Research Reporting Summary linked to this article.

## Data availability

Source data are provided as a Source Data file, and are publicly available at the following repository https://github.com/jaberry/Jimenez_2020 and from the corresponding author upon reasonable request. Source data are provided with this paper.

## Code availability

The codes used for analysis in this manuscript are publicly available at the following repository https://github.com/jaberry/Jimenez_2020 and from the corresponding author upon reasonable request. Source data are provided with this paper.

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

## Acknowledgements

We thank Clay Lacefield, Wei-li Chang, Gergely Turi, Andres Villegas, and Josephine McGowan for pilot work on vCA1 CFC-imaging studies, and Steve Siegelbaum and Fabio Stefanini for comments on the manuscript and scientific discussions. J.C.J. was a Howard Hughes Medical Institute Gilliam Fellow and NIH T32 postdoctoral fellow while conducting this work, and J.C.J., S.C.L., and J.E.B. are Columbia MSTP students. J.E.B. is supported by NIMH: F30 MH117927. J.C.J., S.C.L., J.E.B., S.K.O., and R.H. are supported by NIMH: R37 MH068542, NIA: R01 AG043688, NIMH: R01 MH083862, NYSTEM: NYSTEM-C029157, S10 OD018464, and HDRF: RGA-13-003. M.A.K. is supported by NIMH (R01 MH108623, R01 MH111754, 1R01 MH117961), Weill Scholar Award, IMHRO/One Mind Rising Star Award, Pew Charitable Trusts, and a Klingenstein–Simons Fellowship.

## Author contributions

J.C.J., M.A.K., and R.H. designed the experiments and analysis, J.C.J. and R.H. wrote the paper. J.C.J., S.C.L., and S.K.O. conducted experiments and Ca$^{2+}$ imaging data processing, and J.C.J., S.C.L., and J.E.B. did data analysis. All authors reviewed/revised the manuscript.

## Competing interests

The authors declare no competing interests.
