## [Peer Review File · Nature Communications]

Reviewers' comments:

Reviewer #1 (Remarks to the Author):

Jimenez et al. utilizes a combination of in vivo calcium imaging and optogenetics to understand the mechanisms underlying hippocampal dynamics during memory formation and retrieval. The authors demonstrate that 1.) a greater proportion of BLA- versus LH-projecting ventral hippocampal neurons exhibit footshock responsivity, 2.) shock-responsive neurons fire more synchronously with other shock-nonresponsive cells during fear retrieval, and 3) silencing vHC activity during shock delivery disrupts this enhanced co-activation. The data are very interesting and provide "real-time" evidence, from technically sophisticated manipulations, for the recruitment of synchronized ensembles in the hippocampus during context-evoked fear. In particular, the disruption of correlated firing among ensembles of shock-responsive and unresponsive neurons by training-specific activity suppression has potentially important implications. However, the results are somewhat disjointed and stop well short of establishing either a circuit organization of correlated ensembles or their causal role in context-evoked freezing.

Major comments:

- Graph theory analysis was not adequately described in terms of what information was represented. Also, it seems that ideally the authors would have analyzed correlated units on each day of behavioral testing (e.g. shock-A1, shock-A2, shock-B) to understand the degree to which shock-A2 cell correlations reflect the formation of novel correlated ensembles or could instead be explained by a general increase in coherent activity in the conditioned context. This is important in light of the modulation of local field oscillations by fear memory retrieval and the expected increase in synchrony. The current analysis also does not establish whether there is anything special about shock-responsive cells in terms of their tendency for increased synchrony with other cells during memory retrieval. This is important to establish precisely what is the conceptual advance.
- While the optogenetic interneuron stimulation disrupted the enhanced synchrony between excitatory cells, the lack of any effect on context-evoked freezing makes it difficult to draw the conclusion that this phenomenon represents evidence of memory retrieval, and therefore seems to contradict the main conclusion. Is it assumed that the spared hippocampal hemisphere is compensating? If so, bilateral activation of GABAergic cells should be performed to provide evidence of causal relevance to behavior.
- A model is proposed in which shock responsive vHC cells exhibit coordinated activity with other BLA-projecting neurons (fig 4). Although the authors collected relevant Ca²⁺ imaging data from BLA-projecting neurons that could have been used to test this model, they claim that FOVs from these animals had too few cells to conduct graph theory analysis. It is important to state explicitly how they estimated a priori (i.e. in an unbiased manner) how many cells would be required. Furthermore, if the BLA-projecting dataset cannot be analyzed for correlations, that is fine, but then figure 1 does very little at all to further the conclusions of the paper. We cannot assume anything about projection targets of shock responsive cells in figure 2 based on the observation that proportionately more BLA- than LH-projecting vHC neurons are shock responsive. It may still be the case, for example, that the vast majority of shock responsive vHC cells in figure 2 project to neither BLA nor LH. This makes the current pattern of results of unclear significance.
- The most interesting result in figure 2 is an apparent decrease in the spontaneous activity of LH-projecting cells specifically in the conditioned context. However, this is not addressed in subsequent experiments, nor do the authors comment on the implications of these data, which are intriguing in light of their previous findings that this pathway promotes anxiety-like behavior and does not affect context fear expression (Jimenez et al. Neuron 2018).
- The spatial organization of correlated ensembles should be examined.

Minor comments:

- While it is included in the figure legend and methods, adding some additional information to the figures would be helpful for easy comprehension, for example, such as adding the amount of time between behavioral sessions to Figure 1B & 2A and others, and the n-values to the pie charts in 1E & 2C and others.
- Missing description in figure legend for S3F-H.
- The authors should discuss a potential cellular mechanism that could account for their results. Does interneuron activation disrupt some sort of local plasticity process within CA1 neurons? If so, the relative sparsity of recurrent excitatory connections would seem to be an obstacle. Is there instead a role for network level processing? In this case, it is difficult to understand how this would not interfere with context fear retrieval.

Reviewer #2 (Remarks to the Author):

The manuscript by Jiminez and colleagues examines the role of ventral CA1 (vCA1) neuronal ensembles in contextual fear memory formation and retrieval. In their previous work (Jiminez et al., 2018 Neuron) studying vCA1 projections to the lateral hypothalamus (LHA) and basal amygdala (BA) the authors found that disrupting activity of vCA1-BA but not vCA1-LHA neurons impaired fear memory. In the current set of studies, the authors asked how fear memory representations are encoded in these vCA1-BA neurons. Using 1p-calcium imaging in freely behaving mice, they find that a large proportion of vCA1-BA neurons are strongly activated by foot-shocks during conditioning. Using a separate set of mice expressing GCaMP6 under the synapsin promoter, they construct network graphs based on vCA1 neuron correlations and show that during memory retrieval, vCA1 neurons are more correlated with each other, and more specifically, neurons that were responsive to foot shocks during fear encoding have increased correlated activity with neurons previously not responsive to foot shock (i.e., shock-partner cells). In other words, shock neurons seem to "recruit" shock-partner cells during memory retrieval. To test whether shock-related neuron activity is necessary for the recruitment of shock-partner cells during memory retrieval, they perform simultaneous optogenetics and calcium imaging to inhibit neuronal activity in vCA1 during the training shock presentations. Inhibiting shock-related activity in VGAT::Chrimson mice unilaterally does not produce a deficit in memory, but prevents the recruitment of shock-partner cells in vCA1 during memory retrieval. The authors suggest that this form of network plasticity -- the recruitment of non-shock responsive neurons into highly correlated pairs with shock-cells -- may be necessary for fear memory retrieval. Like their previous work, the current manuscript is technically sophisticated allowing the authors to provide novel insight into how vCA1 circuit dynamics may underlie memory formation and retrieval. However, there are limitations in the network-level analyses and we would suggest additional experiments that we think would strengthen this paper before publication. These are listed below:

Major points

1. The authors argue that shock neurons recruit other neurons to encode a fear memory based on increased numbers of functional correlations that exceed an arbitrary threshold. Although they provide some correction for inflated correlation values for highly active neurons (Fig S2B), the analysis that would convincingly show that there are increased correlated pairs with shock neurons would be to normalize the correlations to shuffled distributions of shock and non-shock cells individually. It is possible that during training and testing sessions shock cells are more active overall than non-shock neurons (outside of freezing bouts), and might therefore contribute to the higher numbers of correlations observed.
2. Moreover, the network graphs are based on neurons present within the FOV on each day, which may change across days. The authors should provide data on what proportion of the total neurons recorded across days are active during each session, and how many neurons are re-activated

across days.

3. The experiment in Fig 3 aims to test the idea that vCA1 activity during the shock periods of training are especially important for later recruitment of shock-partner cells and thus are necessary for fear memory formation.

a. In their experiment to abolish recruitment of shock-partner cells in vCA1, memory retrieval is spared in the VGAT::Chrimson mice, presumably because the increase in correlated activity is spared in the non-manipulated hemisphere. The authors should perform a similar behavioral experiment in which they perform the optogenetic manipulation bilaterally and show that memory is impaired when correlated activity is disrupted in both hemispheres. This experiment is key for the central to the claim that increased correlated activity is necessary for fear memory retrieval.

b. The authors posit that vCA1 activity during the foot shocks is critical for the formation of correlated pairs. However, it may activity outside of these periods (e.g., post-shock periods) that is critical for strengthening these correlations. This is a possibility the authors should attempt to rule out. A group in the behavioral experiment described in the preceding comment in which the optogenetic manipulation is shifted away from the shock period would be sufficient to address this.

4. The statistics for the calcium imaging data are all wilcoxon tests (nonparametric equivalent of paired t-tests) within-groups across A1 and A2. It appears that session B (novel context) is never analyzed and more importantly, between-group comparisons are not made. The effects reported appear robust, so the authors should choose more appropriate statistical tests (e.g., Friedman ANOVAs with appropriate post-hocs).

Reviewer #3 (Remarks to the Author):

In this series of experiments Jimenez and colleagues have used in vivo Ca⁺ imaging in freely behaving mice to examine ventral hippocampal (CA1) cellular activity during the encoding and retrieval of contextual fear conditioning. The main observations are that 1) CA1 neurons projecting to the basal amygdala (BA) but not lateral hypothalamus are shock-responsive, 2) shock-responsive neurons correlate with activity in non-shock responsive neuron pairs during retrieval, 3) unilateral optogenetic inhibition of shock-induced activity during conditioning disrupts the formation of correlated networks during retrieval. The paper is well written and the analyses of the data are rigorous. The technical approach is sophisticated, and imaging activity in defined hippocampal projections during behavior is a strength. However, the relationship between the observed pattern of hippocampal activity and memory encoding and retrieval remains unclear.

1) Although the authors have provided a sophisticated analysis of the hippocampal network activity amongst the imaged ensembles, the transient rates of shock- and non-shock responsive neurons (independent of their correlated activity) across the encoding, retrieval, and generalization tests is not clear. Fig 1C and S1B indicate that there are decreases in firing rate in CA1-LHA neurons during retrieval relative to encoding (and no change in CA1-BA rate). This appears to include all neurons imaged, and spontaneous rate for shock-responsive or shock-non-responsive neurons is not presented. Do shock-responsive-neurons (CA1-BA vs CA1-LHA) change their transient rate from encoding to retrieval sessions? Fig S2G suggests that A2-A1 rates across mice are largely negative (implying that overall spontaneous rates decrease across the sessions), but the population of neurons contributing to that decrease is not clear (BA or LHA projecting? shock-responsive or not?).

2) The authors included a tone-shock manipulation in some mice (Fig S1) that demonstrated that CA1 neurons responsive to shock during context conditioning were also responded to shocks that were preceded by tones. It would be interesting to know whether the neurons were responsive to the tones and whether those tone responses changed as a function of conditioning. Luthi's group (Grewe et al, 2017) has observed non-Hebbian changes in BA calcium transients after auditory fear conditioning, and it would be interesting to know whether those sorts of changes occur in hippocampal neurons projecting to BA. An examination of auditory fear conditioning is beyond the

scope of the current paper, but at minimum the authors should report the responses of hippocampal neurons to the tones, which would provide some information about whether another salient, novel stimulus drives activity in shock-responsive neurons. Of course, recording tone responses in animals without previous history of shock would be a cleaner way to address this question insofar as the tone (and shock) responses recorded during tone-shock trials subsequent to context conditioning may be the result of nonassociative sensitization.

3) The authors claim that correlated activity of shock-responsive neurons (identified during encoding) with non-shock responsive neurons is correlated with "memory strength" (ie, freezing). This correlation, however, does not imply causation and may be driven by the change in behavioral state during retrieval. In other words, fear states may induce correlated network activity in hippocampus (rather than correlated activity driving fear states). The authors suggest that freezing behavior alone does not drive the correlated network activity, because the correlations were higher during periods when the animal was not freezing (FigS2E, F). That said, the overall increase in fear/freezing during retrieval testing may be altering hippocampal transient rate independent of memory.

4) It is not clear whether the neurons described as "shock-responsive" during encoding are not in fact "context+shock" neurons--that is they are firing to the shock US concomitantly with sensory/contextual information. Moreover, it is not clear whether subsequent changes in this population during the retention test is a consequence of associative conditioning (see 3), as opposed to sensitization. The failure to observe correlated activity in the B context during the generalization test suggests that the neural changes are not due to sensitization alone. However, this does not also imply that the changes in the A context during retrieval are associative. The typical control to address these issues is an immediate shock control, which experiences the aversive footshock in the conditioning context, but does not condition to that context. This control would allow one to understand whether hippocampal neurons are responding to the US or are responding to context+shock coincidence that might encode in part an associative memory. It would also allow the authors to determine whether changes in correlated activity during a subsequent retrieval test are due to context memory (as opposed to a delayed consequence of US exposure, or pseudo conditioning). The issues in #3 remain (about fear state rather than memory driving correlations), but at least an immediate shock control would allow a stronger conclusion about whether context memory or mere shock exposure yielded the observations of the authors.

5) The optogenetic inhibition experiment reveals that shock responses during conditioning are causally related to the emergence of correlated CA1 activity during the the retrieval test. This addresses some of the issues raised in #4, but it is not clear whether silencing during the shock or silencing CA1 in general (during a non-shock period) would have yielded a similar outcome. Moreover, this unilateral manipulation did not affect behavioral performance during encoding or retrieval. Hence, it is unclear whether silencing shock-evoked activity impairs memory. Although this experiment addresses some of the concerns raised in #4, the lack of controls for opto inhibition independent of shock and the absence of a bilateral group to show that opto inhibition affects retention limit the conclusions that can be drawn.

We thank the reviewers for their comments, and have revised the manuscript to address their concerns and helpful suggestions as described below.

Reviewer #1:

Jimenez et al. utilizes a combination of in vivo calcium imaging and optogenetics to understand the mechanisms underlying hippocampal dynamics during memory formation and retrieval. The authors demonstrate that 1.) a greater proportion of BLA- versus LH-projecting ventral hippocampal neurons exhibit footshock responsivity, 2.) shock-responsive neurons fire more synchronously with other shock-nonresponsive cells during fear retrieval, and 3) silencing vHC activity during shock delivery disrupts this enhanced co-activation. The data are very interesting and provide “real-time” evidence, from technically sophisticated manipulations, for the recruitment of synchronized ensembles in the hippocampus during context-evoked fear. In particular, the disruption of correlated firing among ensembles of shock-responsive and unresponsive neurons by training-specific activity suppression has potentially important implications. However, the results are somewhat disjointed and stop well short of establishing either a circuit organization of correlated ensembles or their causal role in context-evoked freezing.

Major comments:

1- Graph theory analysis was not adequately described in terms of what information was represented. Also, it seems that ideally the authors would have analyzed correlated units on each day of behavioral testing (e.g. shock-A1, shock-A2, shock-B) to understand the degree to which shock-A2 cell correlations reflect the formation of novel correlated ensembles or could instead be explained by a general increase in coherent activity in the conditioned context. This is important in light of the modulation of local field oscillations by fear memory retrieval and the expected increase in synchrony. The current analysis also does not establish whether there is anything special about shock-responsive cells in terms of their tendency for increased synchrony with other cells during memory retrieval. This is important to establish precisely what is the conceptual advance.

We thank the reviewer for their comment and have added a more detailed description in the manuscript of the information represented by the various correlation graph analyses performed (under results section: “Correlated Activity with vCA1 Shock Cells is Proportional to Memory Strength”):

“...We then utilized these correlation coefficients to conduct graph theory network analysis which can be useful for identifying functional connectivity features^{1,2} which may be essential for executing complex cognitive functions such as associative memory retrieval. Network correlation graphs were generated for each FOV across days by drawing a connection between all correlated pairs (Figure 2B), and graph theory analysis metrics were then computed in order to evaluate the participation of individual neurons in higher-orders of correlated network activity. We focused our analysis on three graph theory metrics, including the number of correlated pairs per neuron, membership of a neuron within a larger inter-connected group (component probability), and the extent to which a given neuron’s pairs were also correlated with each other (clustering coefficient). Analysis of these features would allow us to identify highly interconnected nodes that may enhance network communication (as measured by number of correlated pairs per neuron) and distinct network communities that may restrict the spread of neuronal signals within functional modules (as measured by component probability and clustering coefficient)^{1,2}.”

Moreover, we have included additional analysis to support a specialization of A2 shock-partners to exhibit increased correlated activity during contextual memory retrieval (Fig S5D, see below).

We first assessed the survival of A1 correlated pairs across context conditioning days to assess the degree to which novel correlated ensembles were formed on each imaging day, and found that only ~10% of vCA1 A1 correlated pairs persisted in A2 or B (Fig S5D left panel). This suggests

that vCA1 correlated ensembles undergo a significant degree of reorganization across context conditioning days, which would not be expected if the increased vCA1 correlated activity in A2 simply reflected a general increase in coherent activity in the absence of synaptic plasticity changes.

We next compared the number of new correlated pairs gained across context conditioning days between vCA1 cell types (shock cells, A2 shock-partner, A2 non shock-partner) to assess whether the effect of increased A2 shock-partner correlated activity during A2 retrieval as shown in main Fig 2E-G was a specialized response of those cells during context retrieval only, or a general property of this cell population to exhibit more correlated activity in states of higher network coherence. We thus compared the number of correlated pairs gained during context retrieval A2 to the number of new pairs gained during B tone retrieval (which also resulted in a state of increased vCA1 correlated activity as seen in Fig S11C-E, Btones). We found that A2 shock-partner cells were specialized to gain a significant number of novel correlated pairs during A2 context retrieval, but not during B tone retrieval wherein all 3 cell types exhibited a similar increase in correlated activity (Fig S5D right panel).

Thus, the A2 shock-partner increase in correlated activity in A2 does not reflect a general response property of these cells during any state of increased coherent network activity, but rather is a specific response elicited by A2 context retrieval.

Figure S5. Spatial distribution and remapping of vCA1 correlated cell pairs

d, Left; A minority of correlated pairs on A1 persist in A2 and B, and does not vary by vCA1 cell-type (Kruskal-Wallis by cell type, $N_{\text{shock}}=111$, $N_{\text{A2 shock-partner}}=121$, $N_{\text{A2 non shock-partner}}=616$; A2: $H(2)=3.38$ $p=0.18$, B: $H(2)=0.18$ $p=0.92$). Right; A2 shock-partner cells are specialized to exhibit an increased number of correlated pairs during context retrieval A2 (left), but not to novel context B (middle), or during B tones retrieval (right) (Kruskal-Wallis by cell type, $N_{\text{shock}}=111$, $N_{\text{A2 shock-partner}}=121$, $N_{\text{A2 non shock-partner}}=616$; A2: $H(2)=97.46$ $**p<0.01$, B: $H(2)=0.83$ $p=0.66$; Btones: $H(2)=4.79$ $p=0.09$). Error bars, +/- s.e.m.

Taken together with the interpretation of our graph theory analysis metrics, these data therefore suggest that A2 shock partner cells may serve as highly interconnected nodes that form a network community with shock cells to convey a contextual fear representation. Considering that these highly interconnected nodes are connected with vCA1 shock cells which may be biased to project to the BA (given the enrichment of shock cells within that projection), this functional network may therefore enhance communication of the hippocampal context representation to the BA to drive fear memory retrieval and freezing behavior (see Figure 5, with clarified explanation in the discussion section of manuscript, page 5).

2- While the optogenetic interneuron stimulation disrupted the enhanced synchrony between excitatory cells, the lack of any effect on context-evoked freezing makes it difficult to draw the conclusion that this phenomenon represents evidence of memory retrieval, and therefore seems to contradict the main conclusion. Is it assumed that the spared hippocampal hemisphere is compensating? If so, bilateral activation of GABAergic cells should be performed to provide evidence of causal relevance to behavior.

We agree with the reviewer about the need to provide causal evidence as to the relevance of the vCA1 shock response to behavior, and have therefore added a bilateral vCA1 vGAT-Chrimson

optogenetic experiment in order to disrupt the vCA1 shock response during context encoding and assess the effect on freezing behavior during retrieval (Fig 3, see below). We find that disrupting the vCA1 shock response during context encoding with vGAT-Chrimson stimulation results in a modest but significant decrease in freezing behavior during context retrieval A2 (Fig 3B). This supports the hypothesis that the vCA1 shock response during context encoding is necessary for contextual memory formation, and shock responsive neurons may be essential components of contextual fear memory engrams. When considering that the behavioral effect size of this manipulation was modest (~25% reduction in average % time freezing) relative to the robust Ca²⁺ imaging effects observed in our unilateral optogenetic manipulation during Ca²⁺ imaging (Figure 4), we hypothesize that this reflects a technical limitation in our approach: Considering that partial HPC lesions are insufficient to disrupt contextual memory^{3,4}, optogenetic light delivery through a single fiber per HPC hemisphere may have been insufficient to stimulate vGAT-Chrimson neurons throughout the entire vCA1 volume bilaterally. Therefore, the un-affected HPC regions (which did not receive sufficient optogenetic light delivery) may have partially compensated for the loss-of-function in contextual memory retrieval. In contrast, in our simultaneous Ca²⁺ imaging and unilateral optogenetic silencing manipulation, the optoLED was delivered through the Ca²⁺ imaging lens to manipulate the precise area of HPC from which we detect GCaMP Ca²⁺ signals, therefore allowing us to observe robust effects on correlated neuronal activity.

Figure 3. vCA1 shock activity is necessary for contextual memory encoding

Figure 3. vCA1 shock activity is necessary for contextual memory encoding

a, Experimental design, bilateral vCA1 optogenetic activation of vGAT-Chrimson cells with 532nm laser. Left: DIO-Chrimson or DIO-mCherry was injected into vCA1 of vGAT-Cre mice bilaterally and fiber optics implanted. Right: Example image of vCA1 with expression of Chrimson, and fiber optic targeting. **b**, Left: Experimental design, laser stimulation was delivered for 6 seconds during the footshock in A1 context encoding (ON 178-184 seconds). Right: Bilateral vCA1 shock silencing with vGAT-Chrimson stimulation during context encoding significantly reduced % time freezing during context retrieval relative to vGAT-mCherry control mice (repeated-measures ANOVA; % time freezing*genotype interaction $F_{(1,15)}=6.26$, * $p<0.05$, Chrimson N=8, mCherry N=8). Error bars, +/- s.e.m.

3- A model is proposed in which shock responsive vHC cells exhibit coordinated activity with other BLA-projecting neurons (fig 4). Although the authors collected relevant Ca²⁺ imaging data from BLA-projecting neurons that could have been used to test this model, they claim that FOVs from these animals had too few cells to conduct graph theory analysis. It is important to state explicitly how they estimated a priori (i.e. in an unbiased manner) how many cells would be required.

For all Ca²⁺ imaging datasets (whole-population Syn-GCaMP and projection-specific imaging), we included only those mice with >20 cells per FOV in correlated pair activity analysis based on the following control analyses data (now included in Fig S2 panel I):

Figure S2. Imaging vCA1 correlated activity during contextual fear conditioning

i, The minimum # of cells in an FOV needed to estimate the true correlated activity within a vCA1 cell population was calculated by subsampling different numbers of cells within an FOV (x-axis) and calculating a normalized corr pair ratio (subsampled corr pair ratio/ true ratio; y axis). This normalized corr pair ratio was compared across multiple subsampling iterations, and a 20 cell cut off (dotted line, minimum number of cells sampled for normalized corr pair ratio to approach 1.0 for all iterations) was extrapolated as the minimum number of cells needed in an FOV for an imaging mouse to be included in correlated activity analysis throughout the manuscript. All FOVs with ≥ 50 cells were included in this analysis (N_{mice}=9, N_{cells}=847).

Considering that sparse GCaMP labeling could result in sampling of a vCA1 subpopulation that was not representative of the greater population, we estimated the minimum number of vCA1 subsampled cells needed to accurately re-capitulate the correlated activity of the true population. Subsets of neurons were randomly sampled for multiple iterations from vCA1 mice with FOV's larger than 50, and a subsampling correlated pair ratio was calculated and compared to the whole population correlated pair ratio (Normalized corr pair ratio, y axis). We found that the normalized corr pair ratio approached 1.0 at the 20 cell sampling mark for all sampling iterations tested, and thus applied a 20 cell/ FOV minimum requirement a priori for all vCA1 imaging mice to be included in correlated activity analysis for the current study.

As projection specific cell populations were labeled cells with a retrograde CAV2-Cre viral approach at the downstream target site which results in lower cell labeling yields than local labeling approaches, all but one projection-specific imaging mouse had sparse FOVs with <20 detected GCaMP+ cells (with #cells/FOV ranging from 3-14). Thus, projection-specific imaging mice were not included in correlated activity analysis in the current study. Future studies utilizing whole population GCaMP labeling approaches paired with retrograde opto-tagging techniques may circumvent this technical limitation, and would be critical to further assess the conclusions drawn from the current study.

4- Furthermore, if the BLA-projecting dataset cannot be analyzed for correlations, that is fine, but then figure 1 does very little at all to further the conclusions of the paper. We cannot assume anything about projection targets of shock responsive cells in figure 2 based on the observation that proportionately more BLA- than LH-projecting vHC neurons are shock responsive. It may still be the case, for example, that the vast majority of shock responsive vHC cells in figure 2 project to neither BLA nor LH. This makes the current pattern of results of unclear significance.

We agree with the reviewer that the current study cannot conclude within which vCA1 projection-streams that shock cell correlation networks emerge during contextual fear memory retrieval. While our findings demonstrate that the vCA1-BA projection is enriched in shock responsive

neurons, the findings in the current study may still represent a general property of all vCA1 shock responsive neurons regardless of projection stream. Considering that the majority of shock responsive neurons in our whole-population imaging paradigm become correlated with non-shock cells during A2 retrieval (Fig 2C), future experiments that permit identification of projection-specific neurons within a larger vCA1 GCaMP+ population (possibly utilizing whole-population GCaMP labeling paired with opto-tagging techniques as suggested in the above response) would be necessary to address this.

To reflect the uncertainty of the vCA1-BA shock cell contribution to the findings described in our whole-population imaging analysis, we have added the following comments in the discussion section, page 5:

“Still, in the current study, the participation of vCA1-BA shock cells in functional correlation networks during contextual fear memory retrieval were not examined directly due to the limitations of the retrograde GCaMP labeling techniques employed for imaging Ca²⁺ activity of projection-specific populations. Therefore, future studies examining the activity properties of projection-specific vCA1 populations within the greater vCA1 population are required to support these putative models of vCA1-BA shock cell mediated memory retrieval.”

5- The most interesting result in figure 2 is an apparent decrease in the spontaneous activity of LH-projecting cells specifically in the conditioned context. However, this is not addressed in subsequent experiments, nor do the authors comment on the implications of these data, which are intriguing in light of their previous findings that this pathway promotes anxiety-like behavior and does not affect context fear expression (Jimenez et al. Neuron 2018).

We agree with the reviewer that the finding that the vCA1-LHA projection has decreased activity during contextual fear memory retrieval despite exhibiting increased activity during an innately anxiogenic behavior (as shown in our 2018 Neuron paper) is intriguing. When considering the results of our 2018 Neuron paper within the greater context of HPC function in spatial navigation and contextual memory, we hypothesize that “anxiety-cells” may be specialized to receive entorhinal cortex representations that encode for innately anxiogenic contextual features such as bright lights, open spaces and lack of walls^{5,6}, which are spatial features shared by the EPM open arms and OFT center and utilized in our 2018 Neuron paper analyses. In this model, specialized information routing of these anxiogenic contextual features to subsets of vCA1 neurons that can directly drive avoidance behavior through projections to downstream limbic structures (such as the LHA) may allow for rapid adaptive behavioral responses in real-time. In contrast, in the current study, contextual fear memories are formed by associating a neutral novel context A with an aversive foot shock. The behavioral fear response elicited in A2 requires associative learning and coordination between the HPC-BA⁷, as context A does not elicit an innate fear response during context exploration in A1 (Figure S3B). Thus, vCA1 responses to learned and innately fearful contexts may be segregated into projection-specific populations that are specialized to promote innate behavioral responses such as avoidance (in the LHA pathway) versus associative representations that can then drive learned behavioral responses such as freezing (in the BA pathway).

6- The spatial organization of correlated ensembles should be examined.

We have now included an analysis evaluating the spatial organization of correlated ensembles as suggested (Fig S5A-C, and below). We assessed the distances between cell centroids of all cell pairs (correlated and uncorrelated within an FOV) and find that across all conditioning days and cell types (shock cells, A2 shock-partners, and A2 non shock-partners), cells pairs that have correlated Ca^{2+} activity reside closer to each other within the FOV than non-correlated pairs. This circuit property has recently been reported in other brain structures⁸, and likely reflects local circuit architecture such as shared local inhibitory and/or long range excitatory synaptic inputs between correlated cell pairs.

Figure S5. Spatial distribution and remapping of vCA1 correlated cell pairs

Figure S5. Spatial distribution and reorganization of vCA1 correlated cell pairs

a, Example vCA1 spatial contours of an index cell and its correlated cell pairs across days (A1 top, A2 middle, B bottom panels). Example index cells were selected for each cell type (context-shock cell: left 2 columns, red cells; A2 shock-partner cell: 3rd column from the left, blue cell; A2 non shock-partner: right column, black cell). The red arrow on A2 for the A2 shock-partner cell map indicates the spatial contour of the context-shock cell with which it is correlated. **b**, The centroid distances between vCA1 correlated cell pairs are significantly smaller than distances between non-correlated cell pairs in A1 (left, KStest; KS stat= 0.193 ** $p < 0.01$) and A2 (middle, KStest; KS stat= 0.191 ** $p < 0.01$), but correlated cell pairs have similar distances on A1 and A2 (right, KStest; KS stat= 0.040 $p = 0.21$); ($N_{\text{cells}} = 848$ for all). **c**, Correlated pair cell distances are similar between vCA1 cell types across all context conditioning days (Kruskal-Wallis correlated pair distances for shock, A2 shock-partner, and A2 non shock-partner; A1: correlated pairs per population $N_{\text{shock}} = 123$, $N_{\text{partner}} = 212$, $N_{\text{nonpartner}} = 968$, $H(2) = 1.13$, $p = 0.57$; A2: correlated pairs per population $N_{\text{shock}} = 378$, $N_{\text{partner}} = 406$, $N_{\text{nonpartner}} = 944$, $H(2) = 1.47$, $p = 0.48$; B: correlated pairs per population $N_{\text{shock}} = 135$, $N_{\text{partner}} = 198$, $N_{\text{nonpartner}} = 668$, $H(2) = 1.75$, $p = 0.42$).

Minor comments:

7- While it is included in the figure legend and methods, adding some additional information to the figures would be helpful for easy comprehension, for example, such as adding the amount of time between behavioral sessions to Figure 1B & 2A and others, and the n-values to the pie charts in 1E & 2C and others.

We thank the reviewer for their suggestion and have now added additional labels to figures for easier viewing by our readers (see these updated figures in the main manuscript document).

8- Missing description in figure legend for S3F-H.

The figure legend description for the above panels is now included.

9- The authors should discuss a potential cellular mechanism that could account for their results. Does interneuron activation disrupt some sort of local plasticity process within CA1 neurons? If so, the relative sparsity of recurrent excitatory connections would seem to be an obstacle. Is there instead a role for network level processing? In this case, it is difficult to understand how this would not interfere with context fear retrieval.

We have now expanded our discussion to include a putative cellular mechanism that could account for the disruption in the formation of correlation networks after vCA1 shock-period silencing with vGAT-Chrimson activation (see below). We interpret this effect to be secondary to a loss of proper vCA1 pyramidal neuron activation during the shock period in context encoding (rather than a direct effect of activation within the interneuron population itself). Recent studies have found increased neuronal excitability to promote cell recruitment into a memory trace^{9,10}, as neuronal excitability enhances synaptic plasticity and promotes synaptic strengthening¹¹. We therefore propose a mechanism wherein the shock may result in increased recruitment of vCA1-BA projecting neurons (as this projection is enriched in shock-responsive neurons) to the contextual fear memory engram by increasing vCA1 neuronal excitability at the time of associative memory encoding.

Expanded text in manuscript Discussion, page 5:

“Our work identifies a putative signature of memory retrieval within synchronous ensembles of neurons in vCA1 which is related to memory strength, and incorporates neurons responding to the unconditioned stimulus during training. Moreover, we demonstrate that the vCA1 shock response during context encoding is necessary for the emergence of these functional correlation networks during memory retrieval. This finding may be explained by recent studies which have found that increased neuronal excitability promotes cell recruitment into a memory trace or engram^{9,10}. Therefore, by extension, the shock may result in increased recruitment of vCA1-BA projecting neurons (as this projection is enriched in shock-responsive neurons) to the contextual fear memory engram by increasing neuronal excitability in vCA1 at the time of associative memory encoding.”

Reviewer #2:

The manuscript by Jimenez and colleagues examines the role of ventral CA1 (vCA1) neuronal ensembles in contextual fear memory formation and retrieval. In their previous work (Jimenez et al., 2018 Neuron) studying vCA1 projections to the lateral hypothalamus (LHA) and basal amygdala (BA) the authors found that disrupting activity of vCA1-BA but not vCA1-LHA neurons impaired fear memory. In the current set of studies, the authors asked how fear memory representations are encoded in these vCA1-BA neurons. Using 1p-calcium imaging in freely behaving mice, they find that a large proportion of vCA1-BA neurons are strongly activated by foot-shocks during conditioning. Using a separate set of mice expressing GCaMP6 under the synapsin promoter, they construct network graphs based on vCA1 neuron correlations and show that during memory retrieval, vCA1 neurons are more correlated with each other, and more specifically, neurons that were responsive to foot shocks during fear encoding have increased correlated activity with neurons previously not responsive to foot shock (i.e., shock-partner cells). In other words, shock neurons seem to “recruit” shock-partner cells during memory retrieval. To test whether shock-related neuron activity is necessary for the recruitment of shock-partner cells during memory retrieval, they perform simultaneous optogenetics and calcium imaging to inhibit neuronal activity in vCA1 during the training shock presentations. Inhibiting shock-related activity in VGAT::Chrimson mice unilaterally does not produce a deficit in memory, but prevents the recruitment of shock-partner cells in vCA1 during memory retrieval. The authors suggest that this form of network plasticity -- the recruitment of non-shock responsive neurons into highly correlated pairs with shock-cells -- may be necessary for fear memory retrieval.

Like their previous work, the current manuscript is technically sophisticated allowing the authors to provide novel insight into how vCA1 circuit dynamics may underlie memory formation and retrieval. However, there are limitations in the network-level analyses and we would suggest additional experiments that we think would strengthen this paper before publication. These are listed below:

Major points

1. The authors argue that shock neurons recruit other neurons to encode a fear memory based on increased numbers of functional correlations that exceed an arbitrary threshold. Although they provide some correction for inflated correlation values for highly active neurons (Fig S2B), the analysis that would convincingly show that there are increased correlated pairs with shock neurons would be to normalize the correlations to shuffled distributions of shock and non-shock cells individually.

We thank the reviewer for their comment and have now included a comparison of correlated pairs selected by a threshold determined from a shuffled distribution of shock and non-shock cell activity (to control for potential differences in baseline correlation values secondary to rate differences), and compared that to the Pearson's $R \geq 0.3$ threshold that was used throughout the current study based on ¹². See Fig S6B-D, and below.

We shuffled shock and non-shock cell activity in time (maintaining the total number of Ca^{2+} events per cell across imaging sessions), in order to compute shuffled R values between cell pairs for each shuffle iteration. A cell pair was then considered significantly correlated when the p-value of the true R value was significant relative to the shuffled R distribution (with bonferroni alpha correction based on the number of pair comparisons within the FOV). R values of significant pairs from a shuffled R threshold were then plotted for shock and non-shock cells (Fig S6B bottom panel), and compared to R values of significant pairs as defined by $R \geq 0.3$ threshold (top panel). R values of significant pairs as defined by p-value from shuffled distributions included lower R values for both cell types than the $R \geq 0.3$ threshold (Fig S6B, R values from shuffle distribution were as low as 0.2; difference in R-value range by both thresholds highlighted between panels with gray bars). In line with this difference, we found that defining correlated pair significance by a shuffle distribution resulted in an increased baseline of # of correlated pairs/ cell across all context conditioning days relative to the $R \geq 0.3$ threshold (Fig S6C). Finally, the R value thresholds as defined by significance from the shuffle distribution were not different between shock and non-shock cell types with the exception of A1 (Fig S6B, left bottom panel), which disproportionately increased the number of A1 pairs for shock cells. Moreover, the cell-type specific

effects across conditioning days with the shuffled significance threshold mirrored the results from the $R \geq 0.3$ threshold (Fig S6D), albeit with a shifted baseline reflecting the lower R value threshold.

Taken together, these analyses suggest that the $R \geq 0.3$ threshold used throughout the current study and in (Rajasethupathy et al., 2015) for defining significantly correlated pairs is more stringent than R thresholds determined from shuffled cell activity, and does not disproportionately favor increased correlated activity of shock cells or A2 shock-partners to explain their specialized change in correlated activity during context retrieval.

Figure S6. Increased vCA1 correlated activity during contextual memory retrieval is not driven by cell-type rate differences

b, Cell-type specific comparison of Pearson's R thresholds as defined by $R \geq 0.3$ threshold (top) versus significant p-value differences relative to a shuffled R distribution (bottom); A distribution of shuffled R values was generated for each cell pair by shuffling the timing of cell events and calculating a "shuffled R" for each shuffle iteration (1000 iterations). Significant pairs were then defined by having a true R value that was significantly different from the shuffled distribution (p-value from shuffle with bonferroni correction); Top: Distribution of shock cell and non-shock cell significantly correlated pair R values as defined by $R \geq 0.3$ threshold across context conditioning days (no significant difference between cell populations across days) (KStest non-shock, shock A1: KS stat=0.082 p=0.52, A2: KS stat=0.072 p=0.40, B: KS stat=0.067 p=0.59); Bottom: Distribution of shock cell and non-shock cell significantly correlated pair R values as defined by significant p-value from shuffle across context conditioning days (KStest non-shock, shock A1: KS stat=0.149 **p<0.01, A2: KS stat=0.075 p=0.12, B: KS stat=0.042 p=0.86); Gray boxes highlight the difference in R value range between the distributions in top and bottom panels **c**, The number of correlated pairs per cell is significantly higher across all context conditioning days when defined by p-value from shuffle distribution relative to $R \geq 0.3$ threshold ($N_{\text{cells}}=848$; Mann-Whitney with bonferroni corrected alpha; A1: U=272978.00 **p<0.01, A2: U=285193.50 **p<0.01, B: U=275487.50 **p<0.01). **d**, The same cell-type specific effects across context conditioning days are generated when using $R \geq 0.3$ threshold (left) and significant p-value from shuffle R threshold (right) ($R \geq 0.3$ threshold; $N_{\text{shock}}=111$, $N_{\text{A2 shock-partner}}=121$, $N_{\text{A2 non shock-partner}}=616$; Kruskal-Wallis between groups with bonferroni alpha correction corr pair ratio: A2-A1 **p<0.01 H(2)=49.65, B-A1 p=0.08 H(2)=5.01; wilcoxon sign rank A1,A2 corr pair ratio: shock Z=-2.47 #p<0.05, A2 shock-partner Z=-6.58 ###p<0.01, A2 non shock-partner Z=-0.61 p=0.54 | significant p-value R threshold; $N_{\text{shock}}=111$, $N_{\text{A2 shock-partner}}=162$, $N_{\text{A2 non shock-partner}}=575$; Kruskal-Wallis between groups with bonferroni alpha correction corr pair ratio: A2-A1 **p<0.01 H(2)=59.33, B-A1 p=0.22 H(2)=3.00; wilcoxon sign rank A1,A2 corr pair ratio: shock Z=-2.40 #p<0.05, A2 shock-partner Z=-7.39 ###p<0.01, A2 non shock-partner Z=-0.77 p=0.44).

-It is possible that during training and testing sessions shock cells are more active overall than non-shock neurons (outside of freezing bouts), and might therefore contribute to the higher numbers of correlations observed.

We have now included a Ca^{2+} activity rate comparison of freezing and non-freezing bouts in A2 between cell types (Fig S4D, and below), and find that during A2 context retrieval, shock cells are more active than A2 shock-partner cells or A2 non shock-partners, exclusively during non-freezing bouts (Fig S4D).

Figure S4. vCA1 shock cell correlated activity during memory retrieval

c, Rate of vCA1 subpopulation Ca^{2+} activity across context conditioning days ($N_{shock}=111$, $N_{A2\ shock-partner}=121$, $N_{A2\ non\ shock-partner}=616$; Kruskal-Wallis between groups with bonferroni corrected alpha: A1 $**p<0.01$ $H(2)=26.90$, A2 $**p<0.01$ $H(2)=17.33$, B $p=0.48$ $H(2)=1.48$; wilcoxon sign rank A1,A2; shock: $N_{cells}=111$, $##p<0.01$, $Z=-5.38$; A2 shock-partner: $N_{cells}=121$, $\#p<0.05$, $Z=-2.50$; A2 non shock-partner: $N_{cells}=616$, $##p<0.01$, $Z=-9.30$). **d**, A2 Ca^{2+} activity rate during non-freeze and freeze bouts of vCA1 subpopulations (Kruskal-Wallis between groups with bonferroni corrected alpha: non-freeze $**p<0.01$ $H(2)=20.04$, freeze $p=0.99$ $H(2)=0.03$).

It is important to note that despite the differences in shock cell Ca^{2+} activity rates in A1 and A2 non-freezing bouts, A2 shock-partner cells have similar Ca^{2+} activity rates to A2 non shock-partners (Fig S4C,D), and A2 shock-partner cells have by far the greatest increase in correlated activity during context retrieval (Fig 2H-J). Still, given the difference in shock cell Ca^{2+} activity rates during A1 and A2 non-freezing bouts, we assessed whether the Ca^{2+} activity rates of the individual cell populations during A2 non-freezing bouts could be correlated with the number of A2 correlated pairs, and therefore account for cell-type specific differences in correlated activity during context retrieval (Fig S6A, see below).

We found that the Ca^{2+} activity rate during A2 non-freezing bouts was not correlated with the number of A2 correlated pairs across any cell type (Fig S6A). Thus, cell-type specific Ca^{2+} activity rate differences are unlikely to explain the differences in correlated activity during context retrieval.

Figure S6. Increased vCA1 correlated activity during contextual memory retrieval is not driven by cell-type rate differences or selection bias

a, A2 Ca^{2+} activity rate during non-freeze bouts was not correlated with correlated pair ratio across vCA1 subpopulations ($N_{shock}=111$, $N_{A2\ shock-partner}=121$, $N_{A2\ non\ shock-partner}=616$; Linear regression; A2 non shock-partner: $F_{(1,614)}=0.37$, $p=0.54$, $R^2=0.001$; A2 shock cell: $F_{(1,109)}=2.53$, $p=0.11$, $R^2=0.023$; A2 shock-partner: $F_{(1,119)}=0.06$, $p=0.81$, $R^2<0.001$).

2. Moreover, the network graphs are based on neurons present within the FOV on each day, which may change across days. The authors should provide data on what proportion of the total neurons recorded across days are active during each session, and how many neurons are re-activated across days.

We thank the reviewer for their comment and have now included the suggested analysis in Fig S3C&D (see below). We find that >90% of vCA1 cells are active across context conditioning days and this proportion does not differ between Neutral or Fear contextual conditioning mice (Fig S3C). Similarly, the vast majority of vCA1 neurons (>85%) are reactivated across all 3 days and does not differ by conditioning treatment (Fig S3D).

Figure S3. vCA1 neurons exhibit increased correlated activity during fear memory retrieval

c, The fraction of vCA1 active cells/FOV across conditioning days is similar between Neutral and Fear mice ($N_{\text{Neutral}}=7$, $N_{\text{Fear}}=12$; Mann-Whitney between groups with bonferroni alpha correction; A1: $U=36.00$ $p=0.61$, A2: $U=30.00$ $p=0.31$, B: $U=30.50$ $p=0.33$). **d**, The fraction of re-activated vCA1 cells across context conditioning days is similar between Neutral and Fear mice ($N_{\text{Neutral}}=7$, $N_{\text{Fear}}=12$; Mann-Whitney between groups with bonferroni alpha correction; A1&A2: $U=39.00$ $p=0.80$, A1&B: $U=39.50$ $p=0.83$, A2&B: $U=26.50$ $p=0.19$, all days: $U=40.00$ $p=0.87$).

3. The experiment in Fig 3 aims to test the idea that vCA1 activity during the shock periods of training are especially important for later recruitment of shock-partner cells and thus are necessary for fear memory formation.

a. In their experiment to abolish recruitment of shock-partner cells in vCA1, memory retrieval is spared in the VGAT::Chrimson mice, presumably because the increase in correlated activity is spared in the non-manipulated hemisphere. The authors should perform a similar behavioral experiment in which they perform the optogenetic manipulation bilaterally and show that memory is impaired when correlated activity is disrupted in both hemispheres. This experiment is key for the central to the claim that increased correlated activity is necessary for fear memory retrieval.

We agree with the reviewer and have included this experiment into the current study (Fig 3). We found that bilateral vCA1 vGAT-Chrimson stimulation during the foot shock in A1 context encoding significantly decreased freezing behavior during contextual memory retrieval relative to vGAT-mCherry control mice (see a more detailed description of experiment and data panel above, under Reviewer #1 major comment #2).

b. The authors posit that vCA1 activity during the foot shocks is critical for the formation of correlated pairs. However, it may activity outside of these periods (e.g., post-shock periods) that is critical for strengthening these correlations. This is a possibility the authors should attempt to rule out. A group in the behavioral experiment described in the preceding comment in which the optogenetic manipulation is shifted away from the shock period would be sufficient to address this.

We agree with the reviewer and have now included a light-shifted optogenetic inhibition control experiment in Fig S10 (see below). To assess whether the disruption in correlated activity formation during contextual retrieval is specific to optogenetic inhibition during the A1 shock

period or a general consequence of 6 seconds of vCA1 optogenetic silencing during A1 independent of the shock, we delivered a light-shifted 6 second opto-ON stimulation prior to the shock (mid-session) in a cohort of vGAT-Chrimson control mice. We found that vGAT-Chrimson mediated optogenetic inhibition during the vCA1 shock response (but not during the light-shifted period independent of the shock) during context encoding abolished the formation of correlated activity during context retrieval (Fig S10).

Figure S10. vCA1 shock period silencing disrupts correlated activity during memory retrieval

Figure S10. vCA1 shock period silencing disrupts correlated activity during memory retrieval

a, Experimental design, vGAT-Chrimson mice were imaged in a 3-day contextual fear paradigm, with the optoLED turned ON either during the footshock in A1 to selectively disrupt the vCA1 shock response during context encoding (Footshock silence mice, opto LED ON from 178-184 seconds in A1) or during a Light shifted control period outside of the footshock (Light shift control mice, opto LED ON from 88-94 seconds in A1). **d**, Unilateral shock silencing in A1 did not disrupt freezing behavior during A2 retrieval (repeated-measures ANOVA; % time freezing*genotype interaction $F_{(1,5)}=0.51$, $p=0.62$, $N_{FS-silence}=5$, $N_{Light-shift}=2$). **e**, vCA1-mCherry and vCA1-Chrimson neurons exhibited a significant decrease in Ca²⁺ event rate during context retrieval (A2) (Chrimson $N_{cells}=190$, Light-shift $N_{cells}=69$; Mann-Whitney between groups with bonferroni corrected alpha: A1 $U=5145.50$ $*p<0.01$, A2 $U=6492.00$ $p=0.91$, B $U=3302.50$ $**p<0.01$ | wilcoxon sign rank with bonferroni corrected alpha; Chromson: A1:A2 $Z=-9.13$ $##p<0.01$; Light-shift: A1:A2 $Z=-5.05$ $##p<0.01$). **f-h**, Disrupting the vCA1 shock response during context encoding impaired the formation of increased correlated activity during context retrieval across all correlation graph parameters in vGAT-Chrimson mice, but not in vGAT-Chrimson Light shift control mice (Chrimson $N_{cells}=190$, mCherry $N_{cells}=69$; Mann-Whitney between groups with bonferroni corrected alpha; corr pair ratio A2-A1 $U=3808.50$ $**p<0.01$, B-A1 $U=4944.00$ $**p<0.01$; comp probability A2-A1 $U=4611.00$ $**p<0.01$, B-A1 $U=5193.00$ $**p<0.01$; CC A2-A1 $U=3173.00$ $**p<0.01$, B-A1 $U=4844.00$ $**p<0.01$). Error bars, +/- s.e.m.

4. The statistics for the calcium imaging data are all wilcoxon tests (nonparametric equivalent of paired t-tests) within-groups across A1 and A2. It appears that session B (novel context) is never analyzed and more importantly, between-group comparisons are not made. The effects reported appear robust, so the authors should choose more appropriate statistical tests (e.g., Friedman ANOVAs with appropriate post-hocs).

We have now reported the between-group statistics throughout the manuscript.

Reviewer #3:

In this series of experiments Jimenez and colleagues have used in vivo Ca⁺ imaging in freely behaving mice to examine ventral hippocampal (CA1) cellular activity during the encoding and retrieval of contextual fear conditioning. The main observations are that 1) CA1 neurons projecting to the basal amygdala (BA) but not lateral hypothalamus are shock-responsive, 2) shock-responsive neurons correlate with activity in non-shock responsive neuron pairs during retrieval, 3) unilateral optogenetic inhibition of shock-induced activity during conditioning disrupts the formation of correlated networks during retrieval. The paper is well written and the analyses of the data are rigorous. The technical approach is sophisticated, and imaging activity in defined hippocampal projections during behavior is a strength. However, the relationship between the observed pattern of hippocampal activity and memory encoding and retrieval remains unclear.

1) Although the authors have provided a sophisticated analysis of the hippocampal network activity amongst the imaged ensembles, the transient rates of shock- and non-shock responsive neurons (independent of their correlated activity) across the encoding, retrieval, and generalization tests is not clear. Fig 1C and S1B indicate that there are decreases in firing rate in CA1-LHA neurons during retrieval relative to encoding (and no change in CA1-BA rate). This appears to include all neurons imaged, and spontaneous rate for shock-responsive or shock-non-responsive neurons is not presented. Do shock-responsive-neurons (CA1-BA vs CA1-LHA) change their transient rate from encoding to retrieval sessions? Fig S2G suggests that A2-A1 rates across mice are largely negative (implying that overall spontaneous rates decrease across the sessions), but the population of neurons contributing to that decrease is not clear (BA or LHA projecting? shock-responsive or not?).

We thank the reviewer for their comment and have now included the Ca²⁺ activity rate across days for shock and non-shock cell populations for the projection-specific imaging data (Fig S1B, see below).

We find that shock and non-shock neurons within the vCA1-BA and LHA projection have similar rates in Ca²⁺ transients across context conditioning days (Fig S1B). Thus, vCA1-LHA projecting neurons have a significant decrease in Ca²⁺ transient rate during context retrieval A2 relative to A1 (Fig 1C) which does not vary by shock response (Fig S1B), while the rates of shock and non-shock cells in the vCA1-BA projection do not change across context conditioning days (Fig 1C and S1B).

Figure S1. The vCA1-BA projection has more reliable shock responses than the vCA1-LHA projection

b, vCA1-BA and LHA shock cells have similar Ca²⁺ transient rates across context conditioning days as non shock cells (Mann-Whitney shock vs non shock cells | BA N_{shock}=11 N_{nonshock}=14; A1: U=54.50 p=0.22, A2: U=64.50 p=0.49, B: U=62.50 p=0.43 | LHA N_{shock}=14, N_{nonshock}=54; A1: U=377.00 p=0.99, A2: U=375.00 p=0.96, B: U=354.50 p=0.72).

2) The authors included a tone-shock manipulation in some mice (Fig S1) that demonstrated that CA1 neurons responsive to shock during context conditioning were also responded to shocks that were preceded by tones. It would be interesting to know whether the neurons were responsive to the tones and whether those tone responses changed as a function of conditioning. Luthi's group (Grewe et al, 2017) has observed non-Hebbian changes in BA calcium transients after auditory fear conditioning, and it would be interesting to know whether those sorts of changes occur in hippocampal neurons projecting to

BA. An examination of auditory fear conditioning is beyond the scope of the current paper, but at minimum the authors should report the responses of hippocampal neurons to the tones, which would provide some information about whether another salient, novel stimulus drives activity in shock-responsive neurons. Of course, recording tone responses in animals without previous history of shock would be a cleaner way to address this question insofar as the tone (and shock) responses recorded during tone-shock trials subsequent to context conditioning may be the result of nonassociative sensitization.

We thank the reviewer for their comment and have now included an analysis of shock cell responses to tones, both in our vCA1 projection-specific (Fig S1C-F, see below) and whole-population imaging datasets (S7F-I, see below).

Immediately following 3-minutes of contextual fear retrieval in A2, projection-specific imaging mice received three tone-shock pairings in A2 (Figure S1C). The following day and after 3-minutes of exploration of novel context B, mice were re-exposed to the same tones as the day prior. We found that vCA1-BA and vCA1-LHA shock responsive neurons (responsive to A2 tone-shocks, see behavioral design in S1C below) were not biased to respond to tones in a novel context B (S1E&F).

Figure S1. The vCA1-BA projection has more reliable shock responses than the vCA1-LHA projection

c, Top: Experimental design, vCA1 was imaged over 3 days while mice explored contexts A1, A2, and B for 3 minutes. Mice received a 2-second foot shock at the end of A1, and three 20-second tones that were paired with a 2-second shock at the end of A2. After 3 minutes in neutral context B, mice were exposed to the same three tones again. Bottom: vCA1-BA shock responsive neurons from A1 encoding have a higher rate of Ca²⁺ activity during exposure to tone shocks relative to vCA1-LHA projecting neurons (Mann Whitney NBA=11, NLHA=14, Z=-2.44, P<0.05). **d**, Most vCA1-BA context-shock responsive neurons are also tone-shock responsive neurons while vCA1-LHA shock responsive neurons across shock exposures are significantly less overlapping (Chi squared test of proportions $\chi^2(2)=25.98$, $p<0.0001$ N_{BA}=11, N_{LHA}=40). **e**, vCA1-BA tone-shock cells (from A2, blue in left pie chart and right venn diagram) do not respond to tones in B (yellow in right pie chart and right venn diagram) (N_{tone-shock}=10, N_{tones}=2; random sample overlap control analyses, 2SD range of cell overlap from mock distribution upper= 108.50% lower=26.99%; true overlap=0% Z=1.20, p=0.23). **f**, vCA1-LHA tone-shock cells (from A2, blue in left pie chart and right venn diagram) do not respond to tones in B (yellow in right pie chart and right venn diagram) (N_{tone-shock}=31, N_{tones}=4; random sample overlap control analyses, 2SD range of cell overlap from mock distribution upper= 94.17% lower=3.19%; true overlap=50% Z=0.19, p=0.85). Error bars. +/- s.e.m.

We also assessed the vCA1 shock cell response to tones in a whole-population vCA1 imaging experiment in which mice received tone-shock conditioning in a neutral context A2 (they did not receive a shock in A1), and were exposed to tones in novel context B the following day. We found that vCA1 shock responsive neurons were largely non-overlapping with tone-responsive neurons (FigS7G-I).

Figure S7. vCA1 shock treatment does not elicit correlated activity in the absence of context retrieval, and shock cells are not biased to respond to tones

f, Experimental design, vCA1 was imaged while mice explored neutral contexts A1, A2, and B. After 3 minutes in context A1, mice were exposed to three 20-second tones that were paired with a 2 second shock. After 3 minutes in neutral context B, mice were exposed to the same three tones again. **g**, A subpopulation of vCA1 cells were significantly active to A2 tone-shocks (pie chart, 28% of cells were shock-responsive); Rate of Ca²⁺ transients/sec in A1, A2, and during A2 tone-shocks between shock and non-shock cells ($N_{shock}=30$, $N_{nonshock}=79$). **h**, A subpopulation of vCA1 cells were significantly active to B tones (pie chart, 10% of cells were tone-responsive); Rate of Ca²⁺ transients/sec in A2, B, and during B tones between tone and non-tone cells ($N_{tone}=11$, $N_{nontone}=98$). **i**, Tone-shock cells overlapped with tone cells at chance levels ($N_{tone-shock}=30$, $N_{tones}=11$; random sample overlap control analyses, 2SD range of cell overlap from mock distribution upper= 53.23% lower=1.97%; true overlap=9.1% $Z=-1.44$, $p=0.15$). Error bars, +/- s.e.m.

3) The authors claim that correlated activity of shock-responsive neurons (identified during encoding) with non-shock responsive neurons is correlated with "memory strength" (ie, freezing). This correlation, however, does not imply causation and may be driven by the change in behavioral state during retrieval. In other words, fear states may induce correlated network activity in hippocampus (rather than correlated activity driving fear states). The authors suggest that freezing behavior alone does not drive the correlated network activity, because the correlations were higher during periods when the animal was not freezing (FigS2E, F). That said, the overall increase in fear/freezing during retrieval testing may be altering hippocampal transient rate independent of memory.

We agree with the reviewer about the importance to assess the contribution of freezing behavior alone (independent of memory) to the correlated network activity changes during context retrieval. In addition to the control analysis pointed out by the reviewer that we conducted (correlations are higher during non-freezing bouts), our vCA1 vGAT-Chrimson unilateral shock silencing experiment (Fig 4) provides a controlled condition in which freezing behavior and Ca²⁺ transient event rates are comparable between mCherry and Chrimson groups (Fig 4D&E, see below). Despite the similar % time freezing and Ca²⁺ transient rate exhibited by these two groups, vCA1 correlated activity in A2 context retrieval only emerged in mCherry mice (but not vGAT-Chrimson) in which the vCA1 shock response in A1 was preserved (Fig 4F-H). Thus, this experiment provides a control for behavioral state change after context fear conditioning (wherein freezing

behavior alone in the vGAT-Chrimson group is insufficient to drive increased correlated activity during A2 retrieval), and suggests that the vCA1 shock response in A1 is necessary for the emergence of correlated activity during memory retrieval.

Figure 4. vCA1 shock activity is necessary for correlated activity during memory retrieval

c, Experimental design, vGAT-Chrimson and mCherry control mice were imaged in a 3-day contextual fear paradigm, with the optoLED turned ON during the foot shock in A1 (ON from 178-184 seconds) to selectively disrupt the vCA1 shock response during context encoding. **d**, Unilateral shock silencing in A1 did not disrupt freezing behavior during A2 retrieval (repeated-measures ANOVA; % time freezing*genotype interaction $F_{(1,7)}=1.22$, $p=0.33$, $N_{mCherry}=4$, $N_{Chrimson}=5$). **e**, vCA1-mCherry and vCA1-Chrimson neurons exhibited a similar and significant decrease in Ca²⁺ event rate during context retrieval (A2) (mCherry $N_{cells}=102$, Chrimson $N_{cells}=190$; Mann-Whitney between groups with bonferroni corrected alpha: A1 $U=8636.00$ $p=0.13$, A2 $U=8907.00$ $p=0.26$, B $U=7071.00$ $**p<0.01$ | wilcoxon sign rank with bonferroni corrected alpha; mCherry: A1:A2 $Z=-6.16$ $##p<0.01$, A1:B $Z=-1.329$ $p=0.18$; Chrimson: A1:A2 $Z=-9.13$ $##p<0.01$, B:A1 $Z=-2.16$ $p=0.03$). **f-h**, Disrupting the vCA1 shock response during context encoding impaired the formation of increased correlated activity during context retrieval across all correlation graph parameters in vGAT-Chrimson mice, but not vGAT-mCherry (mCherry $N_{cells}=102$, Chrimson $N_{cells}=190$; Mann-Whitney between groups with bonferroni corrected alpha; corr pair ratio A2-A1 $U=6713.00$ $**p<0.01$, B-A1 $U=8399.00$ $p=0.06$; comp probability A2-A1 $U=6776.00$ $**p<0.01$, B-A1 $U=9217.00$ $p=0.49$; CC A2-A1 $U=6376.00$ $**p<0.01$, B-A1 $U=9025.50$ $p=0.33$). Error bars, +/- s.e.m.

4) It is not clear whether the neurons described as "shock-responsive" during encoding are not in fact "context+shock" neurons--that is they are firing to the shock US concomitantly with sensory/contextual information. Moreover, it is not clear whether subsequent changes in this population during the retention test is a consequence of associative conditioning (see 3), as opposed to sensitization. The failure to observe correlated activity in the B context during the generalization test suggests that the neural changes are not due to sensitization alone. However, this does not also imply that the changes in the A context during retrieval are associative. The typical control to address these issues is an immediate shock control, which experiences the aversive footshock in the conditioning context, but does condition to that context. This control would allow one to understand whether hippocampal neurons are responding to the US or are responding to context+shock coincidence that might encode in part an associative memory. It would also allow the authors to determine whether changes in correlated activity during a subsequent retrieval test are due to context memory (as opposed to a delayed consequence of US exposure, or pseudo conditioning). The issues in #3 remain (about fear state rather than memory driving correlations), but at least an immediate shock control would allow a stronger conclusion about whether context memory or mere shock exposure yielded the observations of the authors.

We thank the reviewer for their comment, and agree that it is unclear from the current study whether “shock-responsive” neurons are not in fact “context+shock” neurons. As footshocks were only delivered in context A in the current study, future studies comparing vCA1 shock responses in contexts with highly dissimilar contextual features will be needed to further elucidate the context specificity of shock responsive neurons. Still, we anticipate a full separation of context+shock representations to be technically challenging given the high level of contextual salience carried by the metallic bars used to deliver the foot shock alone.

As to the reviewer’s comment about correlated activity reflecting a consequence of sensitization from the US exposure alone, we provide the following additional control experiments to further assess for the effect of US shock treatment (independent of contextual memory retrieval) on correlated activity the following day. Our original control for this was to compare correlated activity changes in the conditioned context A2 to a novel context B (day 3 of conditioning). Across all manipulations, we found that vCA1 exhibited increased correlated activity during exposure to the conditioned context A2 but not to a novel context B (Figures 2E-G, 4F-H, and S3F-G). Still, as context exposure A2 and B were not counterbalanced in time following shock exposure in A1 (A2 was conducted at 24 hours after shock exposure while B was conducted at 48 hours), we agree that an immediate shock paradigm or additional control experiments (to further dissociate the effects of contextual memory retrieval and shock treatment on correlated activity) would strengthen the current findings.

To address this, we first conducted pilot immediate shock behavior experiments in sham mice to recapitulate prior studies which found that foot shock delivery at short time delays during exposure to a novel context was insufficient to form a robust context+shock representation^{13,14}. We attempted two different shock delay time points (10 and 20 seconds after A1 session start time), and compared the % time freezing of these groups during A2 retrieval to our 3-minute delay miniscope imaging cohort included throughout the current study (panel A below). We found that both the 10 and 20 second shock delays were sufficient to elicit ~20% time freezing during context retrieval A2 (panel B below). This is in line with other studies which observed ~20% freezing even with delays <20 seconds^{15,16}. The % time freezing elicited by these short shock delay treatments was not significantly different from the % time freezing elicited by the 3-minute delay paradigm used in our imaging mice (panel B), possibly due to differences in behavioral conditions between groups (considering that in our 3-minute delay group, mice are attached to a head-mounted miniscope). Thus, additional pilot studies at even shorter delays, or with more aversive shock treatments at the 3-minute delay time point while under head-mounted miniscope conditions (in order to increase % time freezing for comparison to immediate shock treatments), would be necessary to observe a robust reduction in freezing during context retrieval in the immediate shock paradigm.

Pilot behavior for immediate shock paradigms

Pilot behavior for immediate shock paradigms.

a, Experimental design, mice were placed in context A on day 1 and received a footshock at either a 3 minute, 20-sec, or 10-sec delay. Freezing behavior was then assessed the following day during exposure to the conditioning context A2. **b**, % time freezing was similar between all 3 shock delay groups compared (Kruskal-Wallis $H(2)=0.537$ $p=0.76$; # mice 3min $N=12$, 20sec $N=10$, 10sec $N=10$).

Given the limitation of our immediate shock pilot behavior experiment to dissociate shock and context representations, we next assessed for vCA1 correlated activity changes in a novel context B that was conducted only 24 hours after mice first received a footshock (Fig S7A-E, see below). vCA1 imaging mice were exposed to neutral context A2 (and did not receive a footshock in A1 the day prior), after which they were either exposed to three 20-second tones alone (neutral tones), or three tones paired with a 2-second footshock (tone-shocks) (Figure S7A,B). Mice were then exposed to a novel context B 24 hours later, and the correlated activity across all context conditioning days was compared between groups. We found that correlated activity did not emerge in vCA1 24 hours after shock treatment during exposure to novel context B (Fig S7C-E). Thus, in line with our previous assessment of vCA1 correlated activity in novel context B (48 hours after shock treatment), vCA1 correlated activity does not emerge within 24 hours of shock treatment independent of exposure to the conditioned context.

Figure S7. vCA1 shock treatment does not elicit correlated activity in the absence of context retrieval, and shock cells are not biased to respond to tones

a, Experimental design, vCA1 was imaged while mice explored neutral contexts A1, A2, and B. After 3 minutes in context A2, mice were exposed to three 20-second tones either unpaired (top, Neutral Tones) or paired with a 2-sec shock (bottom, Tone-Shocks). **b**, Freezing behavior across context conditioning days (% Time freezing A2 Neutral Tone mice N=4, A2 Tone-Shock mice N=3). **c-e**, Correlation graph parameters are similar across context conditioning days between mice that received tone-shocks in A2 (orange lines), and those that never received shock treatment (neutral tones, blue lines) (Mann-Whitney between groups with bonferroni corrected alpha; $N_{\text{neutral}}=258$, $N_{\text{tone-shock}}=109$ | corr pair ratio (left); A1: $U=12025.00$ $p=0.03$, A2: $U=13883.00$ $p=0.85$, B: $U=12390.50$ $p=0.07$ | comp memb (middle); A1: $U=13649.50$ $p=0.66$, A2: $U=12265.00$ $p=0.05$, B: $U=13836.50$ $p=0.81$ | CC (right); A1: $U=13742.00$ $p=0.73$, A2: $U=12404.50$ $p=0.07$, B: $U=14044.00$ $p=0.98$).

5) The optogenetic inhibition experiment reveals that shock responses during conditioning are causally related to the emergence of correlated CA1 activity during the the retrieval test. This addresses some of the issues raised in #4, but it is not clear whether silencing during the shock or silencing CA1 in general (during a non-shock period) would have yielded a similar outcome. Moreover, this unilateral manipulation did not affect behavioral performance during encoding or retrieval. Hence, it is unclear whether silencing shock-evoked activity impairs memory. Although this experiment addresses some of the concerns raised in #4, the lack of controls for opto inhibition independent of shock and the absence of a bilateral group to show that opto inhibition affects retention limit the conclusions that can be drawn.

We agree with the reviewer and have included both a bilateral optogenetic vCA1 inhibition experiment into the current study (Fig 3), and a light-shift optogenetic inhibition control experiment, with opto inhibition outside of the shock period during context encoding A1 (Fig S10). We found that bilateral vCA1 vGAT-Chrimson stimulation during the footshock in A1 context encoding significantly decreased freezing behavior during contextual memory retrieval relative to vGAT-mCherry control mice (Fig 3; see a more detailed description of experiment and data panel above, under Reviewer #1 major comment #2). Moreover, optogenetic inhibition during the vCA1 shock response (but not during a light-shifted period independent of the shock) during context encoding abolished the formation of correlated activity during context retrieval (Fig S10; see a more detailed description of experiment and data panel above, under Reviewer #2, comment #3b).

References

1. Bullmore, E. & Sporns, O. Complex brain networks: Graph theoretical analysis of structural and functional systems. *Nature Reviews Neuroscience* (2009). doi:10.1038/nrn2575
2. Sporns, O. Graph theory methods: applications in brain networks. *Dialogues in clinical neuroscience* (2018).
3. Li, H., Matsumoto, K. & Watanabe, H. Different effects of unilateral and bilateral hippocampal lesions in rats on the performance of radial maze and odor-paired associate tasks. *Brain Res. Bull.* (1999). doi:10.1016/S0361-9230(98)00157-9
4. Zhou, H., Zhou, Q. & Xu, L. Unilateral hippocampal inactivation or lesion selectively impairs remote contextual fear memory. *Psychopharmacology (Berl)*. (2016). doi:10.1007/s00213-016-4394-7
5. Lu, L. *et al.* Impaired hippocampal rate coding after lesions of the lateral entorhinal cortex. *Nat Neurosci* **16**, 1085–1093 (2013).
6. Diehl, G. W., Hon, O. J., Leutgeb, S. & Leutgeb, J. K. Grid and Nongrid Cells in Medial Entorhinal Cortex Represent Spatial Location and Environmental Features with Complementary Coding Schemes. *Neuron* (2017). doi:10.1016/j.neuron.2017.03.004
7. Jimenez, J. C. *et al.* Anxiety Cells in a Hippocampal-Hypothalamic Circuit. *Neuron* **97**, 670-683 e6 (2018).
8. Klaus, A. *et al.* The Spatiotemporal Organization of the Striatum Encodes Action Space. *Neuron* **95**, 1171-1180.e7 (2017).
9. Cai, D. J. *et al.* A shared neural ensemble links distinct contextual memories encoded close in time. *Nature* (2016). doi:10.1038/nature17955
10. Yiu, A. P. *et al.* Neurons Are Recruited to a Memory Trace Based on Relative Neuronal Excitability Immediately before Training. *Neuron* (2014). doi:10.1016/j.neuron.2014.07.017
11. Buzsaki, G. Two-stage model of memory trace formation: a role for 'noisy' brain states. *Neuroscience* **31**, 551–570 (1989).
12. Rajasethupathy, P. *et al.* Projections from neocortex mediate top-down control of memory retrieval. *Nature* **526**, 653 (2015).
13. Fanselow, M. S. Factors governing one-trial contextual conditioning. *Anim. Learn. Behav.* (1990). doi:10.3758/BF03205285
14. Rudy, J. W., Barrientos, R. M. & O'Reilly, R. C. Hippocampal formation supports conditioning to memory of a context. *Behav. Neurosci.* (2002). doi:10.1037/0735-7044.116.4.530
15. Frankland, P. W. *et al.* Consolidation of CS and US representations in associative fear conditioning. *Hippocampus* (2004). doi:10.1002/hipo.10208
16. McHugh, T. J. & Tonegawa, S. CA3 NMDA receptors are required for the rapid formation of a salient contextual representation. *Hippocampus* **19**, 1153–1158 (2009).

REVIEWERS' COMMENTS:

Reviewer #1 (Remarks to the Author):

The authors have addressed all of my concerns and the study is now suitable for publishing.

We thank the reviewer for their comments.

Reviewer #2 (Remarks to the Author):

Our comments have all been satisfactorily addressed. Thank you!

We thank the reviewer for their comments.

Reviewer #3 (Remarks to the Author):

The authors have done an outstanding job responding to the reviewer's concerns and I find the paper suitable for publication in its current form.

We thank the reviewer for their comments.